**Data Availability Statement:** All relevant data are available: http://archive.ics.uci.edu/ml.

**Funding:** M. Naderi and A. Bekker acknowledge the research support provided by the National

# A theoretical framework for Landsat data modeling based on the matrix variate mean-mixture of normal model

**Mehrdad Naderi**[1]*, **Andriette Bekker**[1], **Mohammad Arashi**[1,2], **Ahad Jamalizadeh**[3]

**1** Department of Statistics, Faculty of Natural & Agricultural Sciences, University of Pretoria, Pretoria, South Africa, **2** Department of Statistics, Faculty of Mathematical Sciences, Shahrood University of Technology, Shahrood, Iran, **3** Department of Statistics, Faculty of Mathematics and Computer, Shahid Bahonar University of Kerman, Kerman, Iran

* Mehrdad.naderi@ymail.com

## Abstract

This paper introduces a new family of matrix variate distributions based on the mean-mixture of normal (MMN) models. The properties of the new matrix variate family, namely stochastic representation, moments and characteristic function, linear and quadratic forms as well as marginal and conditional distributions are investigated. Three special cases including the restricted skew-normal, exponentiated MMN and the mixed-Weibull MMN matrix variate distributions are presented and studied. Based on the specific presentation of the proposed model, an EM-type algorithm can be directly implemented for obtaining maximum likelihood estimate of the parameters. The usefulness and practical utility of the proposed methodology are illustrated through two conducted simulation studies and through the Landsat satellite dataset analysis.

## 1 Introduction

The skew-normal (SN) distribution, initially introduced by Azzalini [1], has received considerable attention in both theoretical and applied statistics in the past two decades. Various extensions, forms and properties of the SN distribution in the multivariate case were derived in [2–5], and the acknowledged articles therein. An interesting form of the SN distribution was presented by Pyne et al. [3] who named it the restricted multivariate SN (rSN) model. Generally, the rSN distribution can be expressed as a linear transformation of the multivariate normally distributed random vector and the univariate truncated normal distribution. Although the rSN model, like the original SN one, can describe the skewness of data, it still is not robust in dealing with the outlying observations. To cover this drawback, Negarestani et al. [6] used the rSN transformation to introduce the family of multivariate mean mixture of normal (MMN) model. Specifically, a $p$-dimension random vector $\boldsymbol{X}$ is in the family of MMN distributions if

$$\boldsymbol{X} \stackrel{d}{=} \boldsymbol{\mu} + \boldsymbol{\lambda} W + \boldsymbol{Z}, \tag{1}$$

Research Foundation (NRF) of South Africa,
Reference: CPRR160403161466 grant Number:
105840, Reference: SRUG190308422768 grant
Number: 120839 and STATOMET. M. Arashi is also
based upon research supported in part by the NRF
of South Africa, Ref: IFR170227223754 grant
Number: 109214 and SARChI Research Chair-UID:
71199 and Iran National Science Foundation (INSF)
with grant number 97019472.

**Competing interests:** All authors have declared
that no competing interests exist.

where '$\overset{d}{=}$' stands for the equality in distribution, $Z$ follows the multivariate normal model with zero mean and covariance matrix $\Sigma$, and $W$ is an arbitrary random variable independent of $Z$. It is clear that the rSN distribution is a special case of (1) where the mixing variable $W$ is followed by the truncated standard normal distribution lying within a truncated interval $(0, \infty)$, denoted by $W \sim \mathcal{TN}(0, 1; (0, \infty))$. It is shown by Negarestani et al. [6] that the family of MMN may provide a new model with wider range of skewness and kurtosis than the rSN, skew-$t$ [4] and skew Student-$t$-normal [7] distributions. From (1), the probability distribution function (pdf) of random vector $X$ can be presented as

$$f_{\mathrm{MMN}_p}(\boldsymbol{x}; \boldsymbol{\mu}, \boldsymbol{\lambda}, \boldsymbol{\Sigma}, \boldsymbol{v}) = \int_{-\infty}^{\infty} \phi_p(\boldsymbol{x}; \boldsymbol{\mu} + \boldsymbol{\lambda}w, \boldsymbol{\Sigma})h(w; \boldsymbol{v}) \; dw, \quad \boldsymbol{x} \in \mathbb{R}^p, \tag{2}$$

where $\phi_p(\cdot; \cdot)$ denotes the pdf of multivariate normal distribution and $h(\cdot; \boldsymbol{v})$ is the pdf of $W$ parameterized by the vector parameter $\boldsymbol{v}$. The notation $X \sim \mathcal{MMN}_p(\boldsymbol{\mu}, \boldsymbol{\lambda}, \boldsymbol{\Sigma}, W)$ will be used to indicate that $X$ has pdf (2). Depending on the random variable $W$ that can take values on the real line, the pdf (2) can be both symmetric and asymmetric. However, a more flexible and skewed version of the MMN model can be obtain if $W$ has any asymmetric distribution or any positive support model like the truncated-normal, exponential and gamma distributions. Moreover, the pdf (2) can include skew-elliptical models, as the rSN distribution, or can result in skew non-elliptically contoured models if, for example, $W$ is distributed as the exponential, Weibull and gamma models. From Fig 2 in Appendix A, it is observed that the family of MMN distributions offers different orientation compared with the family of mean-variance mixture of normal (MVMN) distributions [8].

Matrix variate distribution finds its genesis in modeling dependent multivariate observations in the normal case [9]. The recent use of the matrix variate normal (MVN) distribution can be found in modeling a wide variety of three-way data appearing in studies including control theory, stochastic systems, image recognition, repeated vector measurements, multivariate time series, spatial data, among others [10, 11]. The MVN distribution not only inherits some appealing properties, features as well as widespread applications from the multivariate normal model, but also it is still not stable and robust against non-normal features such as asymmetry and heavy tails. To deal with the heavy tailed data, Kshirsagar and Bartlett [12] proposed the matrix variate $t$ distribution by showing that the estimator of the parameter matrix of regression coefficients unconditionally follows matrix variate $t$ model. Bulut and Arslan [13] proposed the matrix variate slash distribution as a scale mixture of the matrix variate normal and the uniform distributions. Moreover, in accommodating skewness and kurtosis, the interest of skew distributions provides a platform for robust extension of matrix variate distribution. For instance, works on the matrix variate versions of SN distribution can be found in [14–17]. Even though the matrix variate SN distribution has many attractive properties, it suffers from robustness in dealing with heavy tailed data and from parameter estimation. Regarding these drawbacks of the matrix variate SN model and considering the aforementioned properties of the MMN family of distributions, the objective of this paper is to propose a family of matrix variate mean-mixture of normal (MVMMN) distributions. Some properties and features of our introduced family such as moments, the characteristic function, marginal and conditional distributions are studied. The maximum likelihood (ML) estimate of model parameters are computed by applying expectation-maximization (EM) type algorithm [18].

The contribution of this work can be broken down into six parts. We will begin the usual procedure with the model formulation of the MVMMN distribution in Section 2. Properties and characteristics of the MVMMN distribution are also studied in Section 3. The parameter estimation procedure using the EM-type algorithm and some computational strategies of

implementation are given in Section 4. To examine the performance of the methodology into practice, simulation and real-world data analyses are presented in Sections 5 and 6. Finally, Section 7 gives some concluding remarks and future extensions.

## 2 Proposed family

To start the whole process, we begin with some notations and definitions. A random matrix variable $X \in \mathbb{R}^{p \times n}$ defined as

$$
X = \begin{bmatrix} X_{11} & \cdots & X_{1n} \\ \vdots & \ddots & \vdots \\ X_{p1} & \cdots & X_{pn} \end{bmatrix},
$$

follows a MVN distribution if its pdf is given as

$$
\phi_{p,n}(\boldsymbol{X}; \mathbf{M}, \boldsymbol{\Sigma}, \boldsymbol{\Psi}) = \frac{1}{(2\pi)^{\left(\frac{np}{2}\right)} |\boldsymbol{\Sigma}|^{\frac{n}{2}} |\boldsymbol{\Psi}|^{\frac{p}{2}}} \operatorname{etr}\{-\frac{1}{2}\boldsymbol{\delta}(\boldsymbol{X}, \mathbf{M}, \boldsymbol{\Psi}, \boldsymbol{\Sigma})\},
\tag{3}
$$

where $\operatorname{etr}\{A\} = \exp(\operatorname{tr}(A))$, $\operatorname{tr}(\cdot)$ is the trace operator of a matrix, $\boldsymbol{\delta}(\mathbf{X}, \mathbf{M}, \Psi, \Sigma) = \Sigma^{-1}(\mathbf{X} - \mathbf{M}) \Psi^{-1}(\mathbf{X} - \mathbf{M})^{\top}$ denotes the matrix variate Mahalanobis squared distance, and the mean matrix $\mathbf{M}$ and two dispersion matrices $\boldsymbol{\Sigma} \in \mathbb{R}^{p \times p}$, $\boldsymbol{\Psi} \in \mathbb{R}^{n \times n}$ are defined as

$$
\mathbf{M} = \begin{bmatrix} \mu_{11} & \cdots & \mu_{1n} \\ \vdots & \ddots & \vdots \\ \mu_{p1} & \cdots & \mu_{pn} \end{bmatrix}, \ \boldsymbol{\Sigma} = \begin{bmatrix} \sigma_{11} & \cdots & \sigma_{1p} \\ \vdots & \ddots & \vdots \\ \sigma_{p1} & \cdots & \sigma_{pp} \end{bmatrix}, \ \boldsymbol{\Psi} = \begin{bmatrix} \psi_{11} & \cdots & \psi_{1n} \\ \vdots & \ddots & \vdots \\ \psi_{n1} & \cdots & \psi_{nn} \end{bmatrix}.
$$

We shall use notation $X \sim \mathcal{N}_{p,n}(\mathbf{M}, \boldsymbol{\Sigma}, \boldsymbol{\Psi})$ if $X$ has pdf (3). The following definition is a new result from the representation (1) in the matrix format.

**Definition 1** *A random matrix variable Y is said to have a MVMMN distribution if it can be generated by the stochastic representation*

$$
Y \stackrel{d}{=} \boldsymbol{M} + W\boldsymbol{\Lambda} + X,
\tag{4}
$$

*where $X \sim \mathcal{N}_{p,n}(\boldsymbol{0}, \boldsymbol{\Sigma}, \boldsymbol{\Psi})$, W is a random variable, independent of X, distributed by $h(w; \boldsymbol{\nu})$, and $\boldsymbol{\Lambda} \in \mathbb{R}^{p \times n}$ is a skewness matrix defined as*

$$
\boldsymbol{\Lambda} = \begin{bmatrix} \lambda_{11} & \cdots & \lambda_{1n} \\ \vdots & \ddots & \vdots \\ \lambda_{p1} & \cdots & \lambda_{pn} \end{bmatrix}.
$$

*It can be easily seen that the hierarchical representation of MVMMN model is*

$$
\begin{aligned}
Y|W = w \ &\sim \mathcal{N}_{p,n}(\boldsymbol{M} + w\boldsymbol{\Lambda}, \boldsymbol{\Sigma}, \boldsymbol{\Psi}), \\
W \ &\sim h(w; \boldsymbol{\nu}).
\end{aligned}
\tag{5}
$$

*Hence, the pdf of $Y \sim \mathcal{MVMMN}_{p,n}(\boldsymbol{M}, \boldsymbol{\Lambda}, \boldsymbol{\Sigma}, \boldsymbol{\Psi}, W)$ can be given as*

$$f(\boldsymbol{Y}; \boldsymbol{M}, \boldsymbol{\Lambda}, \boldsymbol{\Sigma}, \boldsymbol{\Psi}, \boldsymbol{v}) = \int_{-\infty}^{\infty} \phi_{p,n}(\boldsymbol{Y}; \boldsymbol{M} + w\boldsymbol{\Lambda}, \boldsymbol{\Sigma}, \boldsymbol{\Psi})h(w; \boldsymbol{v}) \; dw, \qquad \boldsymbol{Y} \in \mathbb{R}^{p \times n}. \tag{6}$$

Applying the well-known property of the MVN distribution, we have

$$Y \sim \mathcal{MVMMN}_{p,n}(\boldsymbol{M}, \boldsymbol{\Lambda}, \boldsymbol{\Sigma}, \boldsymbol{\Psi}, W) \Leftrightarrow \text{vec}(Y) \sim \mathcal{MMN}_{pn}(\text{vec}(\boldsymbol{M}), \text{vec}(\boldsymbol{\Lambda}), \boldsymbol{\Psi} \otimes \boldsymbol{\Sigma}, W), \tag{7}$$

where $\text{vec}(B)$ denotes the vectorization operator of matrix $B$, and $\otimes$ stands for the Kronecker product.

**Remark 1** *Referring to representation (4), it is clear that the mean of Y is $\boldsymbol{M} + \boldsymbol{\Lambda} E(W)$, showing the assumption that the mean of MVMMN distribution is not fixed for all members of the population. We would like to emphasize that the family of matrix variate normal mean-variance mixture (MVNMVM) models [19, 20], assumes that both the mean and variance of the population member are not fixed. Therefore, an interesting extension of the MVMMN distribution can be introduced by considering the family of scale mixture of MVMMN distributions.*

## 2.1 Special cases

- **Restricted matrix variate skew-normal**: If $W \sim \mathcal{TN}(0, 1; (0, \infty))$ in (4), then restricted matrix variate SN (RMVSN) distribution is arisen. The resulting pdf of $Y$ directly obtained by integrating out (6), is

$$f_{\text{RMVSN}}(\boldsymbol{Y}; \boldsymbol{M}, \boldsymbol{\Lambda}, \boldsymbol{\Sigma}, \boldsymbol{\Psi}) = \frac{2}{\boldsymbol{\eta}} \exp\left\{\frac{\mathbf{A}^2}{2}\right\} \phi_{p,n}(\boldsymbol{Y}; \boldsymbol{M}, \boldsymbol{\Sigma}, \boldsymbol{\Psi})\Phi(\mathbf{A}), \qquad \boldsymbol{Y} \in \mathbb{R}^{p \times n}, \tag{8}$$

where $\boldsymbol{\eta}^2 = \text{tr}(\boldsymbol{\Psi}^{-1} \boldsymbol{\Lambda}^\top \boldsymbol{\Sigma}^{-1} \boldsymbol{\Lambda}) + 1$, $\mathbf{A} = \boldsymbol{\eta}^{-1} [\text{tr}(\boldsymbol{\Psi}^{-1} \boldsymbol{\Lambda}^\top \boldsymbol{\Sigma}^{-1}(\boldsymbol{Y} - \boldsymbol{M}))]$, and $\Phi(\cdot)$ denotes the cumulative distribution function of standard normal model.

**Lemma 1** *If $W \sim \mathcal{TN}(\mu, \sigma^2; (0, \infty))$, then*

$$E(W) = \mu + \sigma \frac{\phi(\mu/\sigma)}{\Phi(\mu/\sigma)}, \qquad E(W^r) = \mu E(W^{r-1}) + (r-1)\sigma^2 E(W^{r-2}), \quad r = 2, 3, \ldots,$$

*where $\phi(\cdot)$ is the pdf of standard normal distribution.*

**Proposition 1** *Let $Y \sim \mathcal{RMVSN}(\mathbf{M}, \boldsymbol{\Lambda}, \boldsymbol{\Sigma}, \boldsymbol{\Psi})$ and $W \sim \mathcal{TN}(0, 1; (0, \infty))$. Then, W conditionally on $Y = Y$, denoted by $W_Y$, follows $\mathcal{TN}(\mathbf{A}/\boldsymbol{\eta}, 1/\boldsymbol{\eta}^2; (0, \infty))$.*

**Proof**. Using the hierarchical representation (5), the pdf of RMVSN model (8), and the Bayes' rule, we have

$$f_{W_Y}(w; \mathbf{M}, \boldsymbol{\Lambda}, \boldsymbol{\Sigma}, \boldsymbol{\Psi}) = \frac{2\phi_{p,n}(\mathbf{Y}; \mathbf{M} - w\boldsymbol{\Lambda}, \boldsymbol{\Sigma}, \boldsymbol{\Psi})\phi(w)}{f_{\text{RMVSN}}(\mathbf{Y}; \mathbf{M}, \boldsymbol{\Lambda}, \boldsymbol{\Sigma}, \boldsymbol{\Psi})}$$

$$= \frac{\boldsymbol{\eta} \exp\left\{-0.5(w^2 \left[\text{tr}(\boldsymbol{\Sigma}^{-1}\boldsymbol{\Lambda}\boldsymbol{\Psi}^{-1}\boldsymbol{\Lambda}^\top) + 1\right] - 2w \; \text{tr}(\boldsymbol{\Sigma}^{-1}\boldsymbol{\Lambda}\boldsymbol{\Psi}^{-1}(\mathbf{Y} - \mathbf{M})^\top) + \mathbf{A}^2)\right\}}{\sqrt{2\pi} \; \Phi(\mathbf{A})}$$

$$= \frac{\phi(w; \mathbf{A}/\boldsymbol{\eta}, 1/\boldsymbol{\eta}^2)}{\Phi(\mathbf{A})},$$

which completes the proof after using some matrix factorizations.

- **Convolution with exponential model**: The exponentiated MVMMN (MVMMNE) distribution, say $Y \sim \mathcal{MVMMNE}(\mathbf{M}, \boldsymbol{\Lambda}, \boldsymbol{\Sigma}, \boldsymbol{\Psi})$, is derived as another special case of (4) if $W \sim \mathcal{E}(1)$, where $\mathcal{E}(1)$ denotes the exponential distribution with mean 1. This leads to

obtain the pdf of $Y$ form (6) as

$$f_{\text{MVMMNE}}(\mathbf{Y}; \mathbf{M}, \boldsymbol{\Lambda}, \boldsymbol{\Sigma}, \boldsymbol{\Psi}) = \frac{\sqrt{2\pi}}{\boldsymbol{\eta}_1^*} \exp\left\{\frac{\mathbf{A}_1^{*2}}{2}\right\} \phi_{p,n}(\mathbf{Y}; \mathbf{M}, \boldsymbol{\Sigma}, \boldsymbol{\Psi}) \Phi(\mathbf{A}_1^*), \qquad \mathbf{Y} \in \mathbb{R}^{p \times n},$$

where $\boldsymbol{\eta}_1^{*2} = \boldsymbol{\eta}^2 - 1$, $\mathbf{A}_1^* = \left[\text{tr}(\boldsymbol{\Psi}^{-1}\boldsymbol{\Lambda}^\top\boldsymbol{\Sigma}^{-1}(\mathbf{Y} - \mathbf{M})) - 1\right]/\boldsymbol{\eta}_1^*$.

**Proposition 2** *Let $Y \sim \mathcal{MVMMNE}(\boldsymbol{M}, \boldsymbol{\Lambda}, \boldsymbol{\Sigma}, \boldsymbol{\Psi})$ and $W \sim \mathcal{E}(1)$. Then,*
$W_Y \sim \mathcal{TN}(A_1^*/\boldsymbol{\eta}_1^*, 1/\boldsymbol{\eta}_1^{*2}; (0, \infty))$.

**Proof**. In a similar manner as Proposition 1, the proof can be completed.

- **Convolution with Weibull model**: The mixed-Weibull MVMMN (MVMMNW) distribution, denoted by $Y \sim \mathcal{MVMMNW}(\mathbf{M}, \boldsymbol{\Lambda}, \boldsymbol{\Sigma}, \boldsymbol{\Psi})$, is arisen when $W$ in (4) follows the Weibull model respectively with shape and scale parameters $\alpha = 2$ and $\beta = 1$, $\mathcal{WE}(2, 1)$. Hence, the associated pdf of $Y \sim \mathcal{MVMMNW}(\mathbf{M}, \boldsymbol{\Lambda}, \boldsymbol{\Sigma}, \boldsymbol{\Psi})$ obtained by (6) is

$$f_{\text{MVMMW}}(\mathbf{Y}; \mathbf{M}, \boldsymbol{\Lambda}, \boldsymbol{\Sigma}, \boldsymbol{\Psi}) = \frac{2\sqrt{2\pi}}{\boldsymbol{\eta}_2^{*2}} \exp\left\{\frac{\mathbf{A}_2^{*2}}{2}\right\} \phi_{p,n}(\mathbf{Y}; \mathbf{M}, \boldsymbol{\Sigma}, \boldsymbol{\Psi}) \times \left(\mathbf{A}_2^*\Phi(\mathbf{A}_2^*) + \phi(\mathbf{A}_2^*)\right), \quad \mathbf{Y}$$
$$\in \mathbb{R}^{p \times n},$$

where $\boldsymbol{\eta}_2^{*2} = \boldsymbol{\eta}^2 + 1$, $\mathbf{A}_2^* = \text{tr}(\boldsymbol{\Psi}^{-1}\boldsymbol{\Lambda}^\top\boldsymbol{\Sigma}^{-1}(\mathbf{Y} - \mathbf{M}))/\boldsymbol{\eta}_2^*$.

**Proposition 3** *Let $Y \sim \mathcal{MVMMNW}(\boldsymbol{M}, \boldsymbol{\Lambda}, \boldsymbol{\Sigma}, \boldsymbol{\Psi})$ and $W \sim \mathcal{WE}(2, 1)$. Then, $W_Y$ has the pdf*

$$f_{W_Y}(w; \boldsymbol{M}, \boldsymbol{\Lambda}, \boldsymbol{\Sigma}, \boldsymbol{\Psi}) = \frac{w\,\boldsymbol{\eta}_2^*}{A_2^*\Phi(A_2^*) + \phi(A_2^*)} \phi(w; A_2^*/\boldsymbol{\eta}_2^*, 1/\boldsymbol{\eta}_2^{*2}).$$

*Moreover, for $r = 1, 2, \ldots$,*

$$E(W^r | Y = Y) = \frac{\eta_2^*\Phi(A_2^*)}{A_2^*\Phi(A_2^*) + \phi(A_2^*)} E(V^{r+1}).$$

*where $V \sim \mathcal{TN}(A_2^*/\boldsymbol{\eta}_2^*, 1/\boldsymbol{\eta}_2^{*2}; (0, \infty))$.*

**Proof**. Results can be obtained from the Bayes' rule and some matrix factorizations.

**Theorem 1** *The MVMMN distribution is log-concave if $W$ has log-concave pdf.*

**Proof**. Based on [21], if $f(x)$ and $g(y)$ are log-concave functions, then their convolution, i.e.,

$$\int h(x, y)dy = \int f(x - y)g(y)dy,$$

is also a log-concave function. Hence, the property of vectorization operator of the MVMMN distribution (7) and the fact that the MVN is log-concave completes the proof if $W$ has a log-concave pdf.

**Corollary 1** *The RMVSN, MVMMNE and MVMMNW distributions are log-concave.*

**Proof**. Since the truncated normal, exponential and Weibull (if the shape parameter is $\geq 1$) distributions are log-concave, their associated matrix variate models are, using Theorem 1.

## 3 Characteristics

This section provides some substantial statistical properties of the MVMMN distribution.

**Theorem 2** *If* $Y \sim \mathcal{MVMMN}_{p,n}(M, \Lambda, \Sigma, \Psi, W)$, *then the mean and the characteristic function of* $Y$, *respectively, are*

$$
\begin{aligned}
\mathrm{E}(Y) &= M + \mathrm{E}(W)\Lambda, \\
\varphi_Y(T) &= etr\left\{ iT^\top M - \frac{1}{2} T^\top \Sigma T \Psi \right\} \varphi_W(tr(T^\top \Lambda)), \quad i = \sqrt{-1},
\end{aligned}
$$

*where* $\varphi_W(\cdot)$ *is the characteristic function of* $W \sim h(w; \nu)$.

**Proof.** The proof of theorem can be completed by using the presented representations in Definition 1. Taking expectation on both sides of the stochastic representation (4) the first part is proved. Moreover for the second part, recall that the characteristic function of the matrix variate $X \sim \mathcal{N}_{p,n}(M, \Sigma, \Psi)$ is given as

$$
\varphi_X(T) = etr\left\{ iT^\top M - \frac{1}{2} T^\top \Sigma T \Psi \right\}.
$$

Hence, through the hierarchical representation (5), the characteristic function of $Y$ is obtained by $E(E(\mathrm{tr}(iT^\top Y)|W = w))$.

**Theorem 3** *Let* $Y \sim \mathcal{MVMMN}_{p,n}(M, \Lambda, \Sigma, \Psi, W)$, *and* $M = (m_{ij})$, $\Lambda = (\lambda_{ij})$, $\Sigma = (\sigma_{ij})$, $\Psi = (\psi_{ij})$. *Then, we have*

(i). *For* $1 < i_1, i_2 < p$, *and* $1 < j_1, j_2 < n$,

$$
E(Y_{i_1 j_1} Y_{i_2 j_2}) = \sigma_{i_1 j_1} \sigma_{i_2 j_2} + m_{i_1 j_1} m_{i_2 j_2} + E(W)(m_{i_1 j_1} \lambda_{i_2 j_2} + \lambda_{i_1 j_1} m_{i_2 j_2}) + E(W^2)\lambda_{i_1 j_1} \lambda_{i_2 j_2}.
$$

(ii). *If* $M = 0$,

$$
E(tr(YY^\top)) = E(W^2)\sum_{i=1}^{p}\sum_{j=1}^{n}\lambda_{ij}^2 + tr(\Sigma)tr(\Psi).
$$

**Proof.** (*i*) follows by using the hierarchical representation (5) and applying theorems 2.3.3 of [22]. For $M = 0$, it is clear from part (*i*) that

$$
E(Y_{i_1 j_1} Y_{i_2 j_2}) = \sigma_{i_1 j_1} \sigma_{i_2 j_2} + E(W^2)\lambda_{i_1 j_1} \lambda_{i_2 j_2}.
$$

Therefore, we have

$$
E(tr(YY^\top)) = E\left(\sum_{i=1}^{p}\sum_{j=1}^{n}Y_{ij}Y_{ij}\right) = \sum_{i=1}^{p}\sum_{j=1}^{n}E(Y_{ij}Y_{ij}) = \sum_{i=1}^{p}\sum_{j=1}^{n}\sigma_{ii}\psi_{jj} + E(W^2)\lambda_{ij}\lambda_{ij},
$$

which completes the proof.

**Theorem 4** *The family of MVMMN distributions is closed under the transpose operator, i.e.,*

$$
Y \sim \mathcal{MVMMN}_{p,n}(M, \Lambda, \Sigma, \Psi, W) \leftrightarrow Y^\top \sim \mathcal{MVMMN}_{n,p}(M^\top, \Lambda^\top, \Psi, \Sigma, W).
$$

**Proof.** Based on theorem 2.3.1 of [22], we have

$$
X \sim \mathcal{N}_{p,n}(M, \Sigma, \Psi) \leftrightarrow X^\top \sim \mathcal{N}_{p,n}(M^\top, \Psi, \Sigma).
$$

Now, applying this transpose property of the MVN distribution into the hierarchical representation (5) results in

$$Y|W = w \sim \mathcal{N}_{p,n}(\mathbf{M} + w\mathbf{\Lambda}, \mathbf{\Sigma}, \mathbf{\Psi}) \leftrightarrow Y^\top|W = w \sim \mathcal{N}_{p,n}(\mathbf{M}^\top + w\mathbf{\Lambda}^\top, \mathbf{\Psi}, \mathbf{\Sigma}).$$

**Theorem 5** *Let* $Y \sim \mathcal{MVMMN}_{p,n}(\mathbf{M}, \mathbf{\Lambda}, \mathbf{\Sigma}, \mathbf{\Psi}, W)$, *and* $\mathbf{B}$ *is a* $q \times p$ *matrix of rank* $q \leq p$ *and* $\mathbf{D}$ *is a* $n \times m$ *matrix of rank* $m \leq n$. *Then,*

$$\mathbf{B}Y\mathbf{D} \sim \mathcal{MVMMN}_{q,m}(\mathbf{BMD}, \mathbf{B\Lambda D}, \mathbf{B\Sigma B}^\top, \mathbf{D}^\top\mathbf{\Psi D}, W).$$

**Proof**. The proof of the theorem is completed through obtaining the characteristic function of $\mathbf{B}Y\mathbf{D}$:

$$\varphi_{\mathbf{B}Y\mathbf{D}}(\mathbf{T}) = E(\text{etr}\{i\mathbf{B}Y\mathbf{D}\mathbf{T}^\top\}) = E(\text{etr}\{iY\mathbf{T}_1^\top\}) = \varphi_Y(\mathbf{T}_1^\top),$$

where $\mathbf{T}_1 = \mathbf{D}\mathbf{T}^\top\mathbf{B}$. Now, by applying Theorem 2, we have

$$\varphi_{\mathbf{B}Y\mathbf{D}}(\mathbf{T}) = \text{etr}\left\{i\mathbf{T}^\top\mathbf{BMD} - \frac{1}{2}\mathbf{T}^\top(\mathbf{B\Sigma B}^\top)\mathbf{T}(\mathbf{D}^\top\mathbf{\Psi D})\right\}\varphi_W\big(\text{tr}\big(\mathbf{T}^\top\mathbf{B\Lambda D}\big)\big),$$

which is the characteristic function of $\mathcal{MVMMN}_{q,m}(\mathbf{BMD}, \mathbf{B\Lambda D}, \mathbf{B\Sigma B}^\top, \mathbf{D}^\top\mathbf{\Psi D}, W)$.

**Theorem 6** *Let* $Y \sim \mathcal{MVMMN}_{p,n}(\mathbf{M}, \mathbf{\Lambda}, \mathbf{\Sigma}, \mathbf{\Psi}, W)$, *and partition* $Y$, $\mathbf{M}$, $\mathbf{\Lambda}$, $\mathbf{\Sigma}$, *and* $\mathbf{\Psi}$ *as*

$$Y = \begin{bmatrix} Y_{11} & Y_{12} \\ Y_{21} & Y_{22} \end{bmatrix}, \quad \mathbf{M} = \begin{bmatrix} \mathbf{M}_{11} & \mathbf{M}_{12} \\ \mathbf{M}_{21} & \mathbf{M}_{22} \end{bmatrix}, \quad \mathbf{\Lambda} = \begin{bmatrix} \mathbf{\Lambda}_{11} & \mathbf{\Lambda}_{12} \\ \mathbf{\Lambda}_{21} & \mathbf{\Lambda}_{22} \end{bmatrix},$$

$$\mathbf{\Sigma} = \begin{bmatrix} \mathbf{\Sigma}_{11} & \mathbf{\Sigma}_{12} \\ \mathbf{\Sigma}_{21} & \mathbf{\Sigma}_{22} \end{bmatrix}, \quad \mathbf{\Psi} = \begin{bmatrix} \mathbf{\Psi}_{11} & \mathbf{\Psi}_{12} \\ \mathbf{\Psi}_{21} & \mathbf{\Psi}_{22} \end{bmatrix},$$

*where* $Y_{11}, \mathbf{M}_{11}, \mathbf{\Lambda}_{11} \in \mathbb{R}^{q \times m}, \mathbf{\Sigma}_{11} \in \mathbb{R}^{q \times q}$, *and* $\mathbf{\Psi}_{11} \in \mathbb{R}^{m \times m}$. *Then,*

$$Y_{11} \sim \mathcal{MVMMN}_{q,m}(\mathbf{M}_{11}, \mathbf{\Lambda}_{11}, \mathbf{\Sigma}_{11}, \mathbf{\Psi}_{11}, W).$$

*Similarly, the marginal distribution of* $Y_{12}$, $Y_{21}$, *and* $Y_{22}$ *can be obtained.*

**Proof**. The proof follows by applying Theorem 5 with considering $\mathbf{B} = (\mathbf{I}_q\ \mathbf{0}_{q \times (q-p)})$ and $\mathbf{D} = (\mathbf{I}_m\ \mathbf{0}_{m \times (n-m)})^\top$, where $\mathbf{I}_d$ denotes the unit matrix of order $d$.

**Theorem 7** *Let* $Y \sim \mathcal{MVMMN}_{p,n}(\mathbf{M}, \mathbf{\Lambda}, \mathbf{\Sigma}, \mathbf{\Psi}, W)$, *and partition* $\mathbf{\Psi}$, $\mathbf{\Sigma}$ *as Theorem 6, and* $Y$, $\mathbf{M}$, $\mathbf{\Lambda}$ *as follows*

$$Y = \begin{bmatrix} Y_{1r} \\ Y_{2r} \end{bmatrix} = [Y_{1c}\ Y_{2c}], \quad \mathbf{M} = \begin{bmatrix} \mathbf{M}_{1r} \\ \mathbf{M}_{2r} \end{bmatrix} = [\mathbf{M}_{1c}\ \mathbf{M}_{2c}], \quad \mathbf{\Lambda} = \begin{bmatrix} \mathbf{\Lambda}_{1r} \\ \mathbf{\Lambda}_{2r} \end{bmatrix} = [\mathbf{\Lambda}_{1c}\ \mathbf{\Lambda}_{2c}],$$

*where* $Y_{1r}, \mathbf{M}_{1r}, \mathbf{\Lambda}_{1r} \in \mathbb{R}^{q \times n}$, *and* $Y_{1c}, \mathbf{M}_{1c}, \mathbf{\Lambda}_{1c} \in \mathbb{R}^{p \times m}$. *Then,*

(i). $Y_{1r} \sim \mathcal{MVMMN}_{q,n}(\mathbf{M}_{1r}, \mathbf{\Lambda}_{1r}, \mathbf{\Sigma}_{11}, \mathbf{\Psi}, W)$, *and*
$Y_{1c} \sim \mathcal{MVMMN}_{p,m}(\mathbf{M}_{1c}, \mathbf{\Lambda}_{1c}, \mathbf{\Sigma}, \mathbf{\Psi}_{11}, W)$.

(ii). $Y_{2r}|Y_{1r} = Y_{1r} \sim \mathcal{MVMMN}_{p-q,n}(\mathbf{M}_{2r} + \mathbf{\Sigma}_{21}\mathbf{\Sigma}_{11}^{-1}(Y_{1r} - \mathbf{M}_{1r})$,
$\mathbf{\Lambda}_{2r} - \mathbf{\Sigma}_{21}\mathbf{\Sigma}_{11}^{-1}\mathbf{\Lambda}_{1r}, \mathbf{\Sigma}_{22.1}, \mathbf{\Psi}, W_{Y_{1r}})$, *and* $Y_{2c}|Y_{1c} = Y_{1c} \sim \mathcal{MVMMN}_{p,n-m}$
$(\mathbf{M}_{2c} + (Y_{1c} - \mathbf{M}_{1c})\mathbf{\Psi}_{11}^{-1}\mathbf{\Psi}_{12}, \mathbf{\Lambda}_{2c} - \mathbf{\Lambda}_{1c}\mathbf{\Psi}_{11}^{-1}\mathbf{\Psi}_{12}, \mathbf{\Sigma}, \mathbf{\Psi}_{22.1}, W_{Y_{1c}})$, *where*
$\mathbf{\Sigma}_{22.1} = \mathbf{\Sigma}_{22} - \mathbf{\Sigma}_{21}\mathbf{\Sigma}_{11}^{-1}\mathbf{\Sigma}_{12}, \mathbf{\Psi}_{22.1} = \mathbf{\Psi}_{22} - \mathbf{\Psi}_{21}\mathbf{\Psi}_{11}^{-1}\mathbf{\Psi}_{12}, W_{Y_{1r}} =^d W|Y_{1r} = Y_{1r}$, *and*
$W_{Y_{1c}} =^d W|Y_{1c} = Y_{1c}$.

**Proof**. The proof of (*i*) is completed by considering proper matrices $B$ and $D$ in Theorem 5. Using the hierarchical representation (5) and applying theorem 2.3.12 of [22], the second part of the theorem is proven.

**Corollary 2** *If* $Y \sim \mathcal{RMVSN}(M, \Lambda, \Sigma, \Psi)$ *and under partition of Theorem 7, we have*

(i). $Y_{2r}|Y_{1r} = Y_{1r} \sim \mathcal{MVMMN}_{p-q,n}(M_{2r} + \Sigma_{21}\Sigma_{11}^{-1}(Y_{1r} - M_{1r}), \Lambda_{2r} - \Sigma_{21}\Sigma_{11}^{-1}\Lambda_{1r}, \Sigma_{22.1}, \Psi, W_{Y_{1r}})$ *where* $W_{Y_{1r}} \sim \mathcal{TN}(A_{1r}/\eta_{1r}, 1/\eta_{1r}^2; (0, \infty))$, $\eta_{1r} = tr(\Psi^{-1}\Lambda_{1r}^\top\Sigma_{1r}^{-1}\Lambda_{1r}) + 1$ *and* $A_{1r} = \left[tr(\Psi^{-1}\Lambda_{1r}^\top\Sigma_{1r}^{-1}(Y_{1r} - M_{1r}))\right]/\eta_{1r}$.

(ii). $Y_{2c}|Y_{1c} = Y_{1c} \sim \mathcal{MVMMN}_{p,n-m}(M_{2c} + (Y_{1c} - M_{1c}) \Psi_{11}^{-1}\Psi_{12}, \Lambda_{2c} - \Lambda_{1c}\Psi_{11}^{-1}\Psi_{12}, \Sigma, \Psi_{22.1}, W_{Y_{1c}})$, *where* $W_{Y_{1c}} \sim \mathcal{TN}(A_{1c}/\eta_{1c}, 1/\eta_{1c}^2; (0, \infty))$, $\eta_{1c} = tr(\Psi_{1c}^{-1}\Lambda_{1c}^\top\Sigma^{-1}\Lambda_{1c}) + 1$, *and* $A_{1c} = \left[tr(\Psi_{1c}^{-1}\Lambda_{1c}^\top\Sigma^{-1}(Y_{1c} - M_{1c}))\right]/\eta_{1c}$.

**Corollary 3** *If* $Y \sim \mathcal{MVMMNE}(M, \Lambda, \Sigma, \Psi)$ *and under partition of Theorem (7), we have*

(i). $Y_{2r}|Y_{1r} = Y_{1r} \sim \mathcal{MVMMN}_{p-q,n}(M_{2r} + \Sigma_{21}\Sigma_{11}^{-1}(Y_{1r} - M_{1r}), \Lambda_{2r} - \Sigma_{21}\Sigma_{11}^{-1}\Lambda_{1r}, \Sigma_{22.1}, \Psi, W_{Y_{1r}})$ *where* $W_{Y_{1r}} \sim \mathcal{TN}(A_{1r}^*/\eta_{1r}^*, 1/\eta_{1r}^{*2}; (0, \infty))$, $A_{1r}^* = \left[tr(\Psi^{-1}\Lambda_{1r}^\top\Sigma_{1r}^{-1}(Y_{1r} - M_{1r})) - 1\right]/\eta_{1r}^*$, *and* $\eta_{1r}^* = tr(\Psi^{-1}\Lambda_{1r}^\top\Sigma_{1r}^{-1}\Lambda_{1r})$.

(ii). $Y_{2c}|Y_{1c} = Y_{1c} \sim \mathcal{MVMMN}_{p,n-m}(M_{2c} + (Y_{1c} - M_{1c}) \Psi_{11}^{-1}\Psi_{12}, \Lambda_{2c} - \Lambda_{1c}\Psi_{11}^{-1}\Psi_{12}, \Sigma, \Psi_{22.1}, W_{Y_{1c}})$, *where* $W_{Y_{1c}} \sim \mathcal{TN}(A_{1c}^*/\eta_{1c}^*, 1/\eta_{1c}^{*2}; (0, \infty))$, $A_{1c}^* = \left[tr(\Psi_{1c}^{-1}\Lambda_{1c}^\top\Sigma^{-1}(Y_{1c} - M_{1c})) - 1\right]/\eta_{1c}^*$, *and* $\eta_{1c}^* = tr(\Psi_{1c}^{-1}\Lambda_{1c}^\top\Sigma^{-1}\Lambda_{1c})$.

The presentation of distribution of the matrix quadratic form, done by [23], can also be implemented in the context of the MVMMN family of distributions. Referring to theorem 2.2 of [23], they defined the distribution of quadratic form $\mathcal{Q} = XAX^\top$ to be $\mathcal{Q}_p(A, M, \Sigma)$ where $A$ is a $n \times n$ symmetric real matrix of rank $r$, and $X \sim \mathcal{N}_{p,n}(M, \Sigma, I_n)$.

**Theorem 8** *Let* $Y \sim \mathcal{MVMMN}_{p,n}(M, \Lambda, \Sigma, \Psi, W)$ *and* $W \sim h(w;v)$ *and* $A_{n\times n}$ *any* $n \times n$ *symmetric matrix of rank r. Then, conditionally on* $W = w$,

$$\mathcal{Q} = YAY^\top|W = w \qquad and \qquad B = \sum_{j=1}^{r}\delta_j B_j|W = w,$$

*are identically distributed, where* $\delta_j$ *are the non-zero eigenvalues of* $\Psi^{-\frac{1}{2}}A\Psi^{-\frac{1}{2}}$ *and* $B_j$ *are independent non-central Wishart distribution* $B_j|W = w \sim \mathcal{W}_p(1, \Sigma, m_j m_j^\top)$ *for* $j = 1, \ldots, r$, *where* $m_j = Ma_j$ *and* $a_j$ *are the corresponding orthogonal eigenvectors* $(a_j^\top a_j = 1)$.

**Proof**. Using hierarchical representation (5) of the MVMMN model, we have $Y|W = w \sim \mathcal{N}_{p,n}(M + w\Lambda, \Sigma, \Psi)$. Consequently, the property of the matrix variate normal distribution leads to $Y\Psi^{-\frac{1}{2}}|W = w \sim \mathcal{N}_{p,n}\left(M\Psi^{-\frac{1}{2}} + w\Lambda\Psi^{-\frac{1}{2}}, \Sigma, I\right)$. Now, by definition 2.1 of [23], we have

$$Y\Psi^{-\frac{1}{2}}A\Psi^{-\frac{1}{2}}Y|W = w \sim \mathcal{Q}_p\left(A, M\Psi^{-\frac{1}{2}} + w\Lambda\Psi^{-\frac{1}{2}}, \Sigma\right).$$

On the other hand, through theorem 2.2 of [23], we have

$$\sum_{j=1}^{r}\delta_j B_j|W = w \sim \mathcal{Q}_p\left(A, M\Psi^{-\frac{1}{2}} + w\Lambda\Psi^{-\frac{1}{2}}, \Sigma\right).$$

Therefore, the random matrices $\mathcal{Q}$ and $B$ have identical distributions.

## 4 Parameter estimation

Suppose $N$ matrix observations $\mathbf{Y}_1, \ldots, \mathbf{Y}_N$ of dimension $p \times n$ are drawn independently and identically from the $\mathcal{MVMMN}_{p,n}(\mathbf{M}, \mathbf{\Lambda}, \mathbf{\Sigma}, \mathbf{\Psi}, W)$. Therefore, the log-likelihood function of $\Theta = (\mathbf{M}, \mathbf{\Lambda}, \mathbf{\Sigma}, \mathbf{\Psi}, \nu)$ based on the observed data $\{\mathbf{Y}_i\}_{i=1}^N$ is

$$\ell(\Theta) = \sum_{i=1}^N \log f(\mathbf{Y}_i; \mathbf{M}, \mathbf{\Lambda}, \mathbf{\Sigma}, \mathbf{\Psi}, \nu). \tag{9}$$

To obtain ML estimate of $\Theta$, an EM-type algorithm is implemented as a powerful estimation approach in dealing with the unobserved (missing and/or censored) data and latent variables [18]. The computations of EM algorithm are based on two iterative E- and M-steps. In E-step, the expected value of the complete-data log-likelihood function, the likelihood of the observed and missing data the latent variable, is computed, while in M-step, parameter estimates are updated by maximizing this expected value.

Through the hierarchical representation (5), the complete-data log-likelihood function of $\Theta$, obtained by introducing latent variables $W = (w_1, \ldots, w_N)$ and omitting additive constants, is

$$\begin{aligned}
\ell_c(\Theta) = \quad & \sum_{i=1}^N \log h(w_i; \nu) - \frac{nN}{2} \log |\mathbf{\Sigma}| - \frac{pN}{2} \log |\mathbf{\Psi}| \\
& - \frac{1}{2} \sum_{i=1}^N \{ tr(\boldsymbol{\delta}(\mathbf{Y}_i, \mathbf{M}, \mathbf{\Psi}, \mathbf{\Sigma})) + w_i^2 tr(\mathbf{\Psi}^{-1} \mathbf{\Lambda}^\top \mathbf{\Sigma}^{-1} \mathbf{\Lambda}) \\
& - \omega_i (tr(\mathbf{\Phi}^{-1} \mathbf{\Lambda}^\top \mathbf{\Sigma}^{-1} (\mathbf{Y}_i - \mathbf{M})) + tr(\mathbf{\Phi}^{-1} (\mathbf{Y}_i - \mathbf{M})^\top \mathbf{\Sigma}^{-1} \mathbf{\Lambda})) \}.
\end{aligned} \tag{10}$$

ML estimation of $\Theta$ is performed by using the expectation-conditional maximization (ECM; [24]) algorithm as follows.

- Initialization: Set the number of iteration to $k = 0$ and choose a relative starting point $\Theta^{(k)} = (\mathbf{M}^{(k)}, \mathbf{\Lambda}^{(k)}, \mathbf{\Sigma}^{(k)}, \mathbf{\Psi}^{(k)}, \nu^{(k)})$. We point out that in our data examples the parameters are initialized by $\mathbf{M}^{(0)} = \sum_{i=1}^N \mathbf{Y}_i / N$, $\mathbf{\Lambda}^{(0)} = \mathbf{1}_{p \times n}$, $\mathbf{\Sigma}^{(0)} = c_1 \mathbf{I}_p$, $\mathbf{\Psi}^{(0)} = c_2 \mathbf{I}_n$. Here, $\mathbf{1}_{p \times n}$ is a matrix of dimension $p \times n$ with unit elements. Moreover, the elements of two vectors $c_1$ and $c_2$ are computed, respectively, as

$$c_{1_j} = \sum_{i=1}^N \sum_{l=1}^n (y_{ijl} - \bar{y}_j)^2, \qquad \bar{y}_j = \frac{1}{nN} \sum_{i=1}^N \sum_{l=1}^n y_{ijl}, \qquad j = 1, \ldots, p,$$

$$c_{2_l} = \sum_{i=1}^N \sum_{j=1}^p (y_{ijl} - \bar{y}_l)^2, \qquad \bar{y}_l = \frac{1}{pN} \sum_{i=1}^N \sum_{j=1}^p y_{ijl}, \qquad j = 1, \ldots, n.$$

- E-step: The expected value of the complete-data log-likelihood function (10), called $Q$-function, is computed as

$$\begin{aligned}
Q(\Theta \mid \hat{\Theta}^{(k)}) = \quad & \sum_{i=1}^N \hat{\Upsilon}_i^{(k)} - \frac{nN}{2} \log |\mathbf{\Sigma}| - \frac{pN}{2} \log |\mathbf{\Psi}| \\
& - \frac{1}{2} \sum_{i=1}^N \Big\{ tr(\boldsymbol{\delta}(\mathbf{Y}_i, \mathbf{M}, \mathbf{\Psi}, \mathbf{\Sigma})) + \hat{t}_i^{(k)} tr(\mathbf{\Psi}^{-1} \mathbf{\Lambda}^\top \mathbf{\Sigma}^{-1} \mathbf{\Lambda}) \\
& - \hat{\omega}_i^{(k)} (tr(\mathbf{\Phi}^{-1} \mathbf{\Lambda}^\top \mathbf{\Sigma}^{-1} (\mathbf{Y}_i - \mathbf{M})) + tr(\mathbf{\Phi}^{-1} (\mathbf{Y}_i - \mathbf{M})^\top \mathbf{\Sigma}^{-1} \mathbf{\Lambda})) \Big\}.
\end{aligned} \tag{11}$$

where $\hat{w}_i^{(k)} = E(W|\mathbf{Y}_i, \hat{\Theta}^{(k)})$, $\hat{t}_i^{(k)} = E(W^2|\mathbf{Y}_i, \hat{\Theta}^{(k)})$, and depending on $h(w; \boldsymbol{\nu})$
$\hat{\Upsilon}_i^{(k)} = E(\log h(W; \boldsymbol{\nu})|\mathbf{Y}_i, \hat{\Theta}^{(k)})$.

- First CM-step: Maximizing $Q$-function with respect to M and $\boldsymbol{\Lambda}$ give the following update

$$\hat{\mathbf{M}}^{(k+1)} = \frac{1}{N}\sum_{i=1}^{N}\mathbf{Y}_i - \hat{\boldsymbol{\Lambda}}^{(k)}\bar{w}, \qquad \hat{\boldsymbol{\Lambda}}^{(k+1)} = \frac{N^{-1}\sum_{i=1}^{N}\hat{w}_i^{(k)}\mathbf{Y}_i - N^{-1}\bar{w}\sum_{i=1}^{N}\mathbf{Y}_i}{\bar{t} - \bar{w}^2},$$

where $\bar{w} = N^{-1}\sum_{i=1}^{N}\hat{w}_i^{(k)}$ and $\bar{t} = N^{-1}\sum_{i=1}^{N}\hat{t}_i^{(k)}$.

- Second CM-step: Update $\Sigma$ and $\Psi$, respectively,

$$\begin{aligned}
\hat{\boldsymbol{\Sigma}}^{(k+1)} &= \frac{1}{Nn}\sum_{i=1}^{N}\Big\{\big(\mathbf{Y}_i - \hat{\mathbf{M}}^{(k+1)}\big)(\hat{\boldsymbol{\Psi}}^{(k)})^{-1}(\mathbf{Y}_i - \hat{\mathbf{M}}^{(k+1)})^{\top} \\
&\quad + \hat{t}_i^{(k)}\hat{\boldsymbol{\Lambda}}^{(k+1)}(\hat{\boldsymbol{\Psi}}^{(k)})^{-1}(\hat{\boldsymbol{\Lambda}}^{(k+1)})^{\top} - \hat{w}_i^{(k)}\hat{\boldsymbol{\Lambda}}^{(k+1)}(\hat{\boldsymbol{\Psi}}^{(k)})^{-1}(\mathbf{Y}_i - \hat{\mathbf{M}}^{(k+1)})^{\top} \\
&\quad - \hat{w}_i^{(k)}(\mathbf{Y}_i - \hat{\mathbf{M}}^{(k+1)})(\hat{\boldsymbol{\Psi}}^{(k)})^{-1}(\hat{\boldsymbol{\Lambda}}^{(k+1)})^{\top}\Big\},
\end{aligned}$$

$$\begin{aligned}
\hat{\boldsymbol{\Psi}}^{(k+1)} &= \frac{1}{Np}\sum_{i=1}^{N}\{(\mathbf{Y}_i - \hat{\mathbf{M}}^{(k+1)})^{\top}(\hat{\boldsymbol{\Sigma}}^{(k+1)})^{-1}\big(\mathbf{Y}_i - \hat{\mathbf{M}}^{(k+1)}\big) \\
&\quad + \hat{t}_i^{(k)}(\hat{\boldsymbol{\Lambda}}^{(k+1)})^{\top}(\hat{\boldsymbol{\Sigma}}^{(k+1)})^{-1}\hat{\boldsymbol{\Lambda}}^{(k+1)} \\
&\quad - \hat{w}_i^{(k)}(\hat{\boldsymbol{\Lambda}}^{(k+1)})^{\top}(\hat{\boldsymbol{\Sigma}}^{(k+1)})^{-1}(\mathbf{Y}_i - \hat{\mathbf{M}}^{(k+1)}) \\
&\quad - \hat{w}_i^{(k)}(\mathbf{Y}_i - \hat{\mathbf{M}}^{(k+1)})^{\top}(\hat{\boldsymbol{\Sigma}}^{(k+1)})^{-1}\hat{\boldsymbol{\Lambda}}^{(k+1)}\}.
\end{aligned}$$

- Third CM-step: The additional parameter $\boldsymbol{\nu}$ depending on the distribution of $W_i$ is updated by

$$\hat{\boldsymbol{\nu}}^{(k+1)} = \arg\max_{\boldsymbol{\nu}}\sum_{i=1}^{N}\hat{\Upsilon}_i^{(k)}.$$

**Remark 2** *The conditional expectations $\hat{w}_i$ and $\hat{t}_i$ involved in the Q-function* (11) *can be obtained by Lemma 1 and Propositions 1, 2 and 3 for our three considered models. Furthermore, we note that in all special cases considered in Section 4, the distribution of mixing random variable $W$, is parameter free. Therefore, the last step of the ECM algorithm is not necessary.*

## 4.1 Computational aspects

**4.1.1 Convergence.** The process of the EM algorithm can be iterated until a suitable convergence rule, like $\max \parallel \hat{\Theta}^{(k+1)} - \hat{\Theta}^{(k)} \parallel \le \varepsilon$ or $|\ell(\hat{\Theta}^{(k+1)}) - \ell(\hat{\Theta}^{(k)})| \le \varepsilon$, is satisfied where $\varepsilon$ is a user specified tolerance and $\ell(\cdot)$ is defined in (9). An alternative approach to determine convergence of the EM algorithm is the Aitken acceleration method [25]. To apply this approach, the asymptotic estimate of the log-likelihood at iteration $k + 1$, following [26], can be obtained as

$$\ell_{\infty}(\hat{\Theta}^{(k+1)}) = \ell(\hat{\Theta}^{(k+1)}) + \frac{1}{1 - a^{(k)}}\Big(\ell(\hat{\Theta}^{(k+1)}) - \ell(\hat{\Theta}^{(k)})\Big),$$

where the Aitken acceleration of iteration $k$ is

$$a^{(k)} = \frac{\ell(\hat{\Theta}^{(k+1)}) - \ell(\hat{\Theta}^{(k)})}{\ell(\hat{\Theta}^{(k)}) - \ell(\hat{\Theta}^{(k-1)})}.$$

Therefore, the algorithm can be considered to have converged at iteration $k + 1$ when $\ell_\infty(\hat{\Theta}^{(k+1)}) - \ell(\hat{\Theta}^{(k)}) < \epsilon$, [27]. In our study, the tolerance $\epsilon$ is considered as $10^{-5}$.

**4.1.2 Model selection.** The models in competition in our data analysis are compared using the most commonly used measures Akaike information criterion (AIC; [28]) and Bayesian information criterion (BIC; [29]) defined as

$$\text{AIC} = 2m - 2\ell_{\max} \quad \text{and} \quad \text{BIC} = m \log N - 2\ell_{\max},$$

where $m$ is the number of free parameters and $\ell_{\max}$ is the maximized log-likelihood value. Models with lower values of AIC or BIC are considered more preferable.

# 5 Simulation studies

In this section, the performance of our model and its computational method is illustrated by conducting two simulation studies. The first simulation study aims at comparing the special cases of MVMMN model in dealing with skewed and leptokurtic simulated data. The second simulation study demonstrates whether our proposed ECM algorithm can provide good asymptotic properties.

**Example 1 Model performance**

*In this experiment, simulated data are generated from a matrix variate normal inverse Gaussian (MVNIG; [20]) distribution with sample sizes $N = 50, 100, 500, 1000$ and $2000$, to compare the performance of three special cases of MVMMN model. The MVNIG distribution belongs to the family of MVMVM models where the mixing random variable follows the $\mathcal{GIG}(-0.5, \chi, \psi)$, such that $\mathcal{GIG}$ denotes the generalized inverse Gaussian distribution with parameter $(\kappa, \chi, \psi)$ [30]. We consider this matrix variate distribution to generate non-normal data as it offers the desired level of asymmetry and leptokurtosis. Let $\chi = \psi = 3$ and*

$$\mathbf{M} = \begin{bmatrix} -5 & 2 & 0 & 2 \\ -2 & 0 & 3 & 0 \\ 0 & 1 & 6 & -4 \end{bmatrix}, \quad \mathbf{\Lambda} = \begin{bmatrix} 1 & -1 & 0 & 1 \\ 2 & -1 & 0 & -2 \\ 0 & -1 & 0 & -3 \end{bmatrix},$$

$$\mathbf{\Sigma} = \begin{bmatrix} 1 & -0.5 & 0.1 \\ -0.5 & 1 & 0.5 \\ 0.1 & 0.5 & 1 \end{bmatrix}, \quad \mathbf{\Psi} = \begin{bmatrix} 1 & 0 & 0 & 0 \\ 0 & 1 & -0.5 & 0.5 \\ 0 & -0.5 & 1 & 0.1 \\ 0 & 0.5 & 0.1 & 1 \end{bmatrix}.$$

*Table 1 summarizes the average ($\ell_{AV}$) and standard deviation (Std.) of the maximized log-likelihood together with the frequencies (out of 200 replications) of the particular model chosen based on the biggest $\ell_{max}$ value. The results depicted in Table 1 reveal that the MVMMNE distribution provides a better fit than the other two MVMMN-based models. It is clear that the outperformance of MVMMNE distribution is improved by increasing the sample size, N.*

*In order to compare the accuracy of parameter estimates to the real values, the Frobenius (Frob.) norm is adopted. For a given $d \times m$ matrix $\mathbf{A} = [a_{ij}]$, the Frob. norm is defined as the*

**Table 1. Mean and standard deviation for the maximized log-likelihood and frequency of model outperformance in 200 replications for various sample sizes.**

| | RMVSN | | | MVMMNE | | | MVMMNW | | |
|---|---|---|---|---|---|---|---|---|---|
| $N$ | $\ell_{AV}$ | Std. | Freq. | $\ell_{AV}$ | Std. | Freq. | $\ell_{AV}$ | Std. | Freq. |
| 50 | -731.14 | 35.60 | 55 | -729.90 | 35.54 | 141 | -733.99 | 35.82 | 4 |
| 100 | -1503.40 | 43.96 | 46 | -1501.06 | 43.65 | 154 | -1508.61 | 43.98 | 0 |
| 500 | -7622.68 | 114.60 | 14 | -7610.95 | 113.25 | 186 | -7649.34 | 115.98 | 0 |
| 1000 | -15278.40 | 147.93 | 2 | -15254.32 | 145.88 | 197 | -15329.81 | 148.41 | 1 |
| 2000 | -30574.13 | 206.84 | 1 | -30528.32 | 205.33 | 198 | -30680.56 | 206.61 | 1 |

*square root of the sum of the squares of its elements, i.e.* $||A||_F = \sqrt{\sum_{i=1}^{d} \sum_{j=1}^{m} a_{ij}^2}$. Table 2 *shows the average Frob. norm of* $(\mathbf{M} - \hat{\mathbf{M}}_i), (\mathbf{\Lambda} - \hat{\mathbf{\Lambda}}_i), (\mathbf{\Sigma} - \hat{\mathbf{\Sigma}}_i)$ *and* $(\mathbf{\Psi} - \hat{\mathbf{\Psi}}_i)$, *where* $\hat{\mathbf{M}}_i, \hat{\mathbf{\Lambda}}_i, \hat{\mathbf{\Sigma}}_i$ *and* $\hat{\mathbf{\Psi}}_i$ *are the ML estimates of the fitted model in the ith replication. It is observe that the Frob. norm decreases when the sample size increases. We can also see that the Frob. norm for* $\Sigma$ *and* $\Psi$ *for all models are very close to each other while the MVMME model has the furthest estimates of* $M$ *and* $\Lambda$.

**Example 2 Performance of the model under AR(1) dependent structure**

*In order to investigate the effect of auto-regressive (AR(1)) dependent structure in* $\Sigma$ *and* $\Lambda$ *to the parameter estimates, we conduct another Monte Carlo simulation. In this experiment, we set* $A = \mathbf{0}$ *and* $\Psi^{-1} = I_4$ *and*

$$\mathbf{\Lambda} = \begin{bmatrix} 1 & \lambda & \lambda^2 & \lambda^3 \\ \lambda & 1 & \lambda & \lambda^2 \\ \lambda^2 & \lambda & 1 & \lambda \end{bmatrix} = \left[ \lambda^{|i-j|} \right], \qquad \mathbf{\Sigma} = \begin{bmatrix} 1 & \rho & \rho^2 \\ \rho & 1 & \rho \\ \rho^2 & \rho & 1 \end{bmatrix} = \left[ \rho^{|i-j|} \right],$$

*where* $\lambda = 0.5, 2$ *and* $\rho = 0.5, 0.8$. *For generating a random sample from the MVMMN model, the value 0.001 is added to the diagonal elements of* $\Sigma$ *to ensure that it is a positive definite matrix.*

*In each replication of 200 trials, the we generate data from the MVNIG distribution with true parameter values displayed above and* $\chi = \psi = 3$ *for the sample sizes N = 100 and 1000. By fitting the RMVSN, MVMMNE and MVMMNW distributions to the generated data, the Frob. norm of* $(\mathbf{M} - \hat{\mathbf{M}}), (\mathbf{\Lambda} - \hat{\mathbf{\Lambda}}), (\mathbf{\Sigma} - \hat{\mathbf{\Sigma}})$ *and* $(\mathbf{\Psi} - \hat{\mathbf{\Psi}})$ *are obtained.* Table 3 *summarizes the average Frob.*

**Table 2. Mean of Frob. norm for parameter estimates of the candidate distributions for various sample sizes.**

| parameter→ | $\mathbf{M} - \hat{\mathbf{M}}$ | | | $\mathbf{\Lambda} - \hat{\mathbf{\Lambda}}$ | | |
|---|---|---|---|---|---|---|
| $N\downarrow$ | RMVSN | MVMMNE | MVMMNW | RMVSN | MVMMNE | MVMMNW |
| 50 | 1.7135 | 2.0228 | 1.1425 | 1.3717 | 2.1097 | 1.5439 |
| 100 | 1.6714 | 2.0197 | 0.8540 | 1.0385 | 2.0324 | 1.0986 |
| 500 | 1.5130 | 1.8976 | 0.3955 | 0.6317 | 1.9081 | 0.6722 |
| 1000 | 1.4822 | 1.8772 | 0.3033 | 0.5559 | 1.8806 | 0.6163 |
| 2000 | 1.4796 | 1.864 1 | 0.2460 | 0.5268 | 1.8752 | 0.5931 |
| parameter→ | $\mathbf{\Sigma} - \hat{\mathbf{\Sigma}}$ | | | $\mathbf{\Psi} - \hat{\mathbf{\Psi}}$ | | |
| $N\downarrow$ | RMVSN | MVMMNE | MVMMNW | RMVSN | MVMMNE | MVMMNW |
| 50 | 0.3536 | 0.3546 | 0.3520 | 0.4378 | 0.4370 | 0.4361 |
| 100 | 0.2311 | 0.2314 | 0.2300 | 0.2997 | 0.3000 | 0.2986 |
| 500 | 0.1043 | 0.1039 | 0.1044 | 0.1300 | 0.1291 | 0.1303 |
| 1000 | 0.0744 | 0.0739 | 0.0741 | 0.0981 | 0.0976 | 0.0979 |
| 2000 | 0.0515 | 0.0511 | 0.0513 | 0.0707 | 0.0701 | 0.0702 |

**Table 3. Mean of Frob. norm for parameter estimates of the candidate distributions for some selected values of $\lambda$ and $\rho$.**

| | | $\mathbf{M} - \hat{\mathbf{M}}$ | | | | | | $\mathbf{\Lambda} - \hat{\mathbf{\Lambda}}$ | | | | | |
| | | RMVSN | | MVMMNE | | MVMMNW | | RMVSN | | MVMMNE | | MVMMNW | |
| $\lambda$ | $\rho$ | 100 | 1000 | 100 | 1000 | 100 | 1000 | 100 | 1000 | 100 | 1000 | 100 | 1000 |
| 0.5 | 0.5 | 1.1167 | 0.5698 | 1.2671 | 0.9176 | 1.2919 | 0.4306 | 1.2240 | 0.3709 | 1.2753 | 0.9257 | 1.5625 | 0.6880 |
| | 0.8 | 1.0401 | 0.6036 | 1.2095 | 0.9326 | 1.1386 | 0.3661 | 1.1023 | 0.3522 | 1.2184 | 0.9407 | 1.3944 | 0.6036 |
| 2 | 0.5 | 3.5017 | 2.9992 | 4.1028 | 3.5957 | 1.2340 | 0.7310 | 1.8416 | 1.3392 | 4.2938 | 3.7512 | 1.3229 | 0.6188 |
| | 0.8 | 3.3566 | 1.1863 | 3.8494 | 1.4686 | 1.2609 | 0.3373 | 1.8779 | 0.6136 | 4.0888 | 1.5914 | 1.1736 | 0.2349 |
| | | $\mathbf{\Sigma} - \hat{\mathbf{\Sigma}}$ | | | | | | $\mathbf{\Psi} - \hat{\mathbf{\Psi}}$ | | | | | |
| | | RMVSN | | MVMMNE | | MVMMNW | | RMVSN | | MVMMNE | | MVMMNW | |
| $\lambda$ | $\rho$ | 100 | 1000 | 100 | 1000 | 100 | 1000 | 100 | 1000 | 100 | 1000 | 100 | 1000 |
| 0.5 | 0.5 | 0.2829 | 0.1004 | 0.2754 | 0.0963 | 0.2821 | 0.0943 | 0.3376 | 0.1028 | 0.3319 | 0.1026 | 0.3384 | 0.1000 |
| | 0.8 | 0.2931 | 0.0916 | 0.2889 | 0.0903 | 0.2906 | 0.0890 | 0.3345 | 0.0995 | 0.3310 | 0.1002 | 0.3333 | 0.0979 |
| 2 | 0.5 | 0.2412 | 0.0768 | 0.2395 | 0.0747 | 0.2411 | 0.0779 | 0.3041 | 0.0917 | 0.3048 | 0.0931 | 0.3039 | 0.0912 |
| | 0.8 | 0.2489 | 0.0470 | 0.2463 | 0.0464 | 0.2480 | 0.0477 | 0.3064 | 0.0536 | 0.3070 | 0.0550 | 0.3056 | 0.0532 |

*norm of the ML estimates of the fitted models. As expected, the Frob. norm of the parameters decreases as the sample size increases. It can also be observed that the MVMMNW distribution has the smallest Frob. norm of $(\mathbf{M} - \hat{\mathbf{M}})$ and $(\mathbf{\Lambda} - \hat{\mathbf{\Lambda}})$ for the selected combinations of $\lambda$ and $\rho$.*

**Example 3 Finite sample properties of the ML estimates**

*The second simulation study aims at investigating the finite-sample properties of ML estimators obtained by using the ECM algorithm. We consider the situation where Monte Carlo samples of sizes N = 100 and 500 are generated for each of the three special cases of MVMMN distribution. The presumed parameters for all distributions are same as used in Example 1. Fig 1 shows the marginal distributions of the columns, labeled by V1, V2, V3, and V4, for the RMVSN, MVMMNE and MVMMNW distributions of a typical dataset with size 100. The solid red line highlights the marginal mean. In each replication of 1000 trials, the synthetic dataset was fitted with the true generator model via ECM algorithm. To investigate the estimation accuracies, we calculate the bias and the mean squared error (MSE), defined as*

$$\text{Bias} = \frac{1}{1000} \sum_{k=1}^{1000} \hat{\theta}_k - \theta_{true} \quad and \quad \text{MSE} = \frac{1}{1000} \sum_{k=1}^{1000} (\hat{\theta}_k - \theta_{true})^2,$$

*where $\hat{\theta}_k$ denotes the ML estimate of $\theta_{true}$ (a specific parameter) at the kth replication.*

*The detailed numerical results are reported in Table 4. It can be observed that the bias and MSE for all three special cases of MVMMN distribution tend to decrease toward zero by increasing the sample size, showing empirically the consistency of the ML estimates obtained via the ECM algorithm.*

## 6 Analysis of Landsat data

To investigate the performance of the developed model in real-world data analysis, we consider Landsat satellite data (LSD) originally obtained by NASA and available at Irvine machine learning repository (http://archive.ics.uci.edu/ml). Each line of the LSD contains of four spectral values of nine pixel neighborhoods in a satellite image. In other words, the lines of LSD are related to a matrix of observations of $4 \times 9$ dimension. Moreover, each of the LSD matrix of observations belongs to one of six different classes, namely red soil, cotton crop, grey soil,

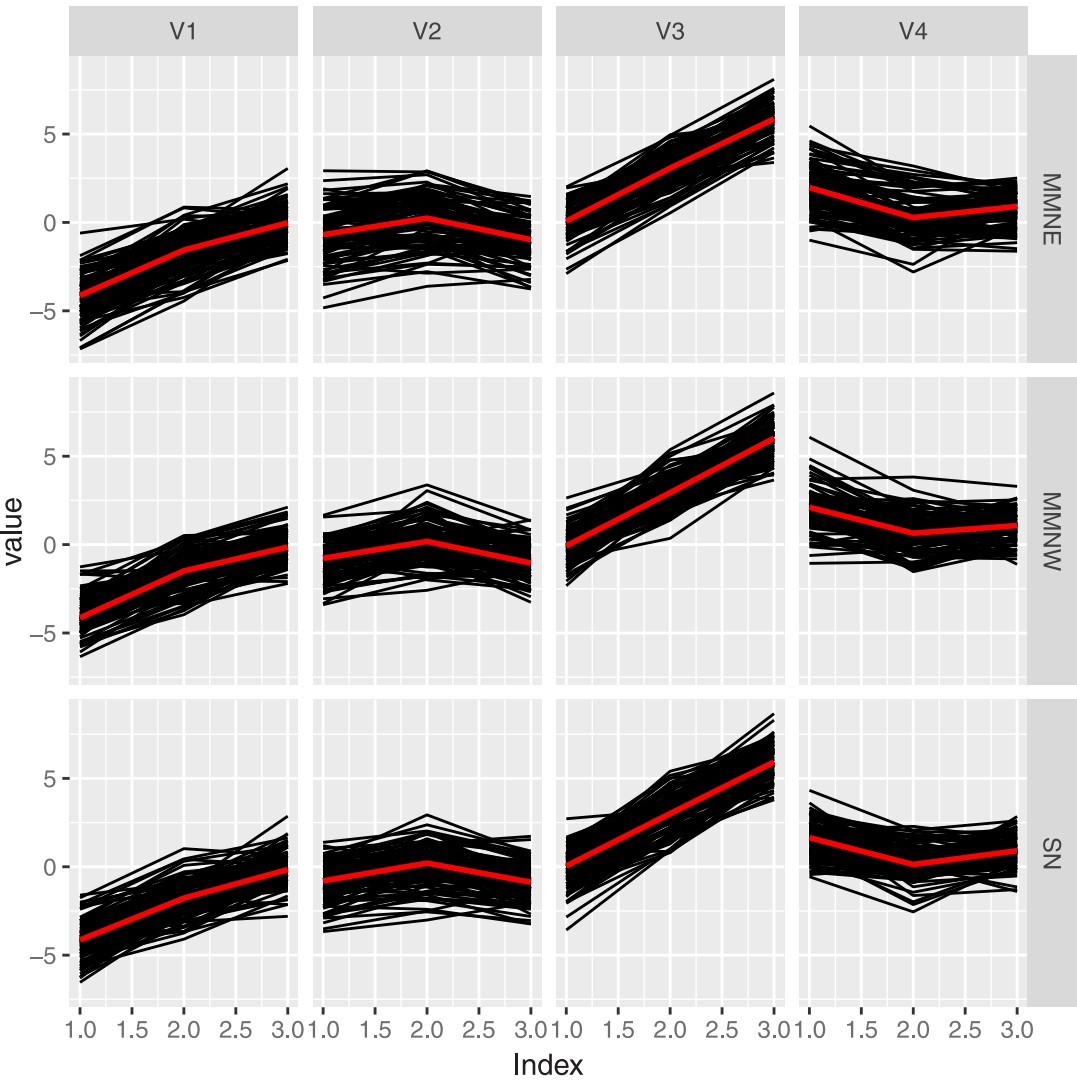

**Fig 1. Marginals of a typical simulated data form the RMVSN, MVMMNE and MVMMNW distributions if the drawing has been lengthwise stretched.**

damp grey soil, soil with vegetation stubble, and very damp grey soil. In our analysis, we focus on two classes, the red soil and cotton crop, with size 461 and 224, respectively, for illustrative purposes.

We fitted RMMVSN, MVMMNE and MVMMNW distributions by implementing the ECM algorithm. Table 5 shows a summary of ML fitting results, including the parameter estimates, maximized log-likelihood values, AIC and BIC of the three fitted models. It is observed that the MVMMNW and MVMMNE distributions respectively outperform the others for the red soil and cotton crop data. Based on the values of the shape matrix $\Lambda$, it is clear that the estimated skewness parameters are moderately to highly significant, showing that the distribution of matrix observation is skewed. Moreover, the estimated scale matrices $\Sigma$ and $\Psi$ highlight the covariance structure in the data.

**Table 4. Simulation results for assessing the consistency of ML parameter estimates with two sample sizes.**

| Model | N | Measure | M | Λ | Σ | Ψ |
|---|---|---|---|---|---|---|
| RMVSN | 100 | Bias | $\begin{bmatrix} 0.015 & 0.003 & -0.002 & 0.015 \\ -0.008 & -0.014 & 0.005 & -0.018 \\ -0.007 & -0.007 & 0.002 & -0.010 \end{bmatrix}$ | $\begin{bmatrix} -0.019 & -0.001 & 0.001 & -0.015 \\ 0.012 & 0.014 & -0.009 & 0.015 \\ 0.009 & 0.006 & -0.007 & 0.008 \end{bmatrix}$ | $\begin{bmatrix} -0.023 & 0.010 & -0.003 \\ 0.010 & -0.024 & -0.014 \\ -0.003 & -0.014 & -0.025 \end{bmatrix}$ | $\begin{bmatrix} -0.001 & -0.002 & 0.004 & -0.002 \\ -0.002 & 0.007 & -0.003 & 0.004 \\ 0.004 & -0.003 & 0.003 & -0.002 \\ -0.002 & 0.004 & -0.002 & 0.003 \end{bmatrix}$ |
| | | MSE | $\begin{bmatrix} 0.030 & 0.030 & 0.030 & 0.028 \\ 0.035 & 0.030 & 0.031 & 0.037 \\ 0.030 & 0.029 & 0.030 & 0.042 \end{bmatrix}$ | $\begin{bmatrix} 0.036 & 0.036 & 0.030 & 0.034 \\ 0.055 & 0.037 & 0.031 & 0.055 \\ 0.029 & 0.035 & 0.031 & 0.083 \end{bmatrix}$ | $\begin{bmatrix} 0.006 & 0.003 & 0.003 \\ 0.003 & 0.006 & 0.004 \\ 0.003 & 0.004 & 0.007 \end{bmatrix}$ | $\begin{bmatrix} 0.007 & 0.004 & 0.004 & 0.005 \\ 0.004 & 0.006 & 0.004 & 0.005 \\ 0.004 & 0.004 & 0.005 & 0.004 \\ 0.005 & 0.005 & 0.004 & 0.007 \end{bmatrix}$ |
| | 500 | Bias | $\begin{bmatrix} 0.004 & 0.001 & -0.001 & 0.001 \\ -0.002 & 0.001 & -0.001 & -0.003 \\ 0.000 & 0.004 & 0.000 & -0.001 \end{bmatrix}$ | $\begin{bmatrix} -0.005 & 0.000 & 0.000 & 0.000 \\ 0.004 & 0.000 & 0.000 & 0.001 \\ 0.000 & -0.004 & 0.003 & -0.001 \end{bmatrix}$ | $\begin{bmatrix} -0.004 & 0.002 & 0.001 \\ 0.002 & -0.004 & -0.002 \\ 0.001 & -0.002 & -0.003 \end{bmatrix}$ | $\begin{bmatrix} -0.001 & -0.001 & 0.001 & 0.000 \\ -0.001 & 0.002 & -0.001 & 0.003 \\ 0.001 & -0.001 & 0.000 & -0.001 \\ 0.000 & 0.003 & -0.001 & 0.002 \end{bmatrix}$ |
| | | MSE | $\begin{bmatrix} 0.006 & 0.006 & 0.006 & 0.006 \\ 0.007 & 0.006 & 0.006 & 0.007 \\ 0.006 & 0.006 & 0.006 & 0.009 \end{bmatrix}$ | $\begin{bmatrix} 0.007 & 0.007 & 0.006 & 0.007 \\ 0.011 & 0.007 & 0.006 & 0.010 \\ 0.006 & 0.007 & 0.006 & 0.016 \end{bmatrix}$ | $\begin{bmatrix} 0.001 & 0.001 & 0.001 \\ 0.001 & 0.001 & 0.001 \\ 0.001 & 0.001 & 0.001 \end{bmatrix}$ | $\begin{bmatrix} 0.001 & 0.001 & 0.001 & 0.001 \\ 0.001 & 0.001 & 0.001 & 0.001 \\ 0.001 & 0.001 & 0.001 & 0.001 \\ 0.001 & 0.001 & 0.001 & 0.001 \end{bmatrix}$ |
| MVMMNE | 100 | Bias | $\begin{bmatrix} 0.015 & -0.010 & -0.004 & 0.004 \\ 0.011 & 0.000 & 0.003 & -0.009 \\ 0.005 & 0.003 & -0.002 & -0.016 \end{bmatrix}$ | $\begin{bmatrix} -0.022 & 0.021 & 0.002 & -0.021 \\ -0.039 & 0.018 & -0.003 & 0.036 \\ 0.002 & 0.016 & -0.003 & 0.057 \end{bmatrix}$ | $\begin{bmatrix} -0.017 & 0.009 & 0.004 \\ 0.009 & -0.017 & -0.011 \\ 0.004 & -0.011 & -0.018 \end{bmatrix}$ | $\begin{bmatrix} 0.001 & -0.003 & 0.005 & 0.000 \\ -0.003 & 0.008 & -0.006 & 0.003 \\ 0.005 & -0.006 & 0.005 & -0.003 \\ 0.000 & 0.003 & -0.003 & -0.001 \end{bmatrix}$ |
| | | MSE | $\begin{bmatrix} 0.021 & 0.020 & 0.020 & 0.019 \\ 0.023 & 0.020 & 0.019 & 0.024 \\ 0.020 & 0.018 & 0.018 & 0.030 \end{bmatrix}$ | $\begin{bmatrix} 0.020 & 0.020 & 0.011 & 0.018 \\ 0.047 & 0.019 & 0.011 & 0.045 \\ 0.011 & 0.018 & 0.011 & 0.092 \end{bmatrix}$ | $\begin{bmatrix} 0.005 & 0.003 & 0.003 \\ 0.003 & 0.006 & 0.004 \\ 0.003 & 0.004 & 0.006 \end{bmatrix}$ | $\begin{bmatrix} 0.006 & 0.004 & 0.004 & 0.004 \\ 0.004 & 0.006 & 0.005 & 0.004 \\ 0.004 & 0.005 & 0.006 & 0.003 \\ 0.004 & 0.004 & 0.003 & 0.006 \end{bmatrix}$ |
| | 500 | Bias | $\begin{bmatrix} 0.000 & 0.000 & -0.003 & 0.001 \\ 0.003 & 0.000 & 0.002 & 0.002 \\ 0.001 & 0.002 & -0.002 & 0.003 \end{bmatrix}$ | $\begin{bmatrix} -0.004 & 0.002 & 0.002 & -0.005 \\ -0.007 & 0.004 & -0.001 & 0.007 \\ 0.002 & 0.002 & 0.003 & 0.011 \end{bmatrix}$ | $\begin{bmatrix} -0.002 & 0.001 & 0.001 \\ 0.001 & -0.003 & -0.003 \\ 0.001 & -0.003 & -0.003 \end{bmatrix}$ | $\begin{bmatrix} -0.001 & -0.002 & 0.002 & 0.000 \\ -0.002 & 0.003 & -0.002 & 0.001 \\ 0.002 & -0.002 & 0.002 & -0.001 \\ 0.000 & 0.001 & -0.001 & -0.001 \end{bmatrix}$ |
| | | MSE | $\begin{bmatrix} 0.003 & 0.002 & 0.002 & 0.003 \\ 0.003 & 0.003 & 0.003 & 0.003 \\ 0.002 & 0.003 & 0.002 & 0.003 \end{bmatrix}$ | $\begin{bmatrix} 0.003 & 0.003 & 0.002 & 0.002 \\ 0.004 & 0.003 & 0.002 & 0.004 \\ 0.002 & 0.002 & 0.002 & 0.007 \end{bmatrix}$ | $\begin{bmatrix} 0.001 & 0.001 & 0.001 \\ 0.001 & 0.001 & 0.001 \\ 0.001 & 0.001 & 0.001 \end{bmatrix}$ | $\begin{bmatrix} 0.001 & 0.001 & 0.001 & 0.001 \\ 0.001 & 0.001 & 0.001 & 0.001 \\ 0.001 & 0.001 & 0.001 & 0.001 \\ 0.001 & 0.001 & 0.001 & 0.001 \end{bmatrix}$ |
| MVMMNW | 100 | Bias | $\begin{bmatrix} 0.005 & -0.010 & 0.001 & 0.004 \\ 0.008 & 0.001 & -0.003 & -0.015 \\ 0.011 & -0.007 & 0.004 & -0.021 \end{bmatrix}$ | $\begin{bmatrix} -0.006 & 0.014 & -0.007 & -0.006 \\ -0.009 & 0.003 & 0.005 & 0.022 \\ -0.006 & 0.012 & -0.003 & 0.028 \end{bmatrix}$ | $\begin{bmatrix} -0.023 & 0.011 & -0.004 \\ 0.011 & -0.023 & -0.014 \\ -0.004 & -0.014 & -0.030 \end{bmatrix}$ | $\begin{bmatrix} 0.002 & -0.001 & 0.003 & -0.001 \\ -0.001 & 0.006 & -0.004 & 0.004 \\ 0.003 & -0.004 & 0.003 & -0.002 \\ -0.001 & 0.004 & -0.002 & 0.003 \end{bmatrix}$ |
| | | MSE | $\begin{bmatrix} 0.053 & 0.056 & 0.050 & 0.056 \\ 0.062 & 0.055 & 0.050 & 0.067 \\ 0.053 & 0.058 & 0.055 & 0.085 \end{bmatrix}$ | $\begin{bmatrix} 0.056 & 0.061 & 0.051 & 0.058 \\ 0.071 & 0.058 & 0.051 & 0.073 \\ 0.057 & 0.063 & 0.055 & 0.104 \end{bmatrix}$ | $\begin{bmatrix} 0.006 & 0.004 & 0.003 \\ 0.004 & 0.006 & 0.004 \\ 0.003 & 0.004 & 0.007 \end{bmatrix}$ | $\begin{bmatrix} 0.007 & 0.005 & 0.004 & 0.005 \\ 0.005 & 0.006 & 0.005 & 0.005 \\ 0.004 & 0.005 & 0.006 & 0.004 \\ 0.005 & 0.005 & 0.004 & 0.008 \end{bmatrix}$ |
| | 500 | Bias | $\begin{bmatrix} -0.004 & 0.001 & 0.001 & 0.002 \\ 0.003 & 0.000 & -0.001 & -0.006 \\ 0.001 & -0.001 & 0.002 & -0.003 \end{bmatrix}$ | $\begin{bmatrix} 0.002 & 0.004 & -0.003 & -0.006 \\ -0.007 & 0.001 & 0.003 & 0.010 \\ 0.000 & 0.002 & -0.003 & 0.006 \end{bmatrix}$ | $\begin{bmatrix} -0.007 & 0.003 & 0.000 \\ 0.003 & -0.004 & -0.002 \\ 0.000 & -0.002 & -0.005 \end{bmatrix}$ | $\begin{bmatrix} 0.002 & -0.001 & 0.002 & -0.001 \\ -0.001 & 0.001 & -0.001 & 0.001 \\ 0.002 & -0.001 & 0.000 & -0.002 \\ -0.001 & 0.001 & -0.002 & 0.000 \end{bmatrix}$ |
| | | MSE | $\begin{bmatrix} 0.011 & 0.011 & 0.01 & 0.011 \\ 0.013 & 0.011 & 0.01 & 0.013 \\ 0.010 & 0.011 & 0.01 & 0.017 \end{bmatrix}$ | $\begin{bmatrix} 0.012 & 0.011 & 0.01 & 0.012 \\ 0.014 & 0.011 & 0.01 & 0.014 \\ 0.010 & 0.012 & 0.01 & 0.020 \end{bmatrix}$ | $\begin{bmatrix} 0.001 & 0.001 & 0.001 \\ 0.001 & 0.001 & 0.001 \\ 0.001 & 0.001 & 0.001 \end{bmatrix}$ | $\begin{bmatrix} 0.001 & 0.001 & 0.001 & 0.001 \\ 0.001 & 0.001 & 0.001 & 0.001 \\ 0.001 & 0.001 & 0.001 & 0.001 \\ 0.001 & 0.001 & 0.001 & 0.001 \end{bmatrix}$ |

**Table 5. Parameters estimates and the performance summary of three matrix models on the LSD subsets.**

| Dataset | Parameter | red soil | cotton crop |
|---|---|---|---|
| MVRSN | M | $\begin{bmatrix} 54.50 & 53.70 & 53.88 & 53.63 & 53.39 & 53.75 & 53.64 & 53.88 & 54.03 \\ 76.02 & 75.05 & 75.05 & 74.40 & 73.96 & 74.09 & 73.97 & 73.95 & 74.11 \\ 91.92 & 91.21 & 91.46 & 90.98 & 89.88 & 90.31 & 90.76 & 90.44 & 89.65 \\ 77.10 & 76.85 & 77.29 & 76.28 & 75.78 & 76.27 & 76.65 & 76.13 & 76.04 \end{bmatrix}$ | $\begin{bmatrix} 46.00 & 42.45 & 40.70 & 48.54 & 45.92 & 43.23 & 50.43 & 49.26 & 47.54 \\ 34.02 & 28.35 & 25.29 & 38.63 & 33.85 & 29.33 & 42.76 & 40.14 & 36.56 \\ 115.13 & 116.99 & 116.38 & 113.82 & 115.33 & 116.39 & 111.31 & 112.05 & 112.91 \\ 123.46 & 128.03 & 127.46 & 119.52 & 124.22 & 126.47 & 114.94 & 116.70 & 119.61 \end{bmatrix}$ |
| | $\Lambda$ | $\begin{bmatrix} 11.01 & 11.62 & 10.98 & 11.51 & 11.46 & 10.70 & 10.77 & 10.32 & 10.24 \\ 22.97 & 23.97 & 23.05 & 24.49 & 25.17 & 24.09 & 23.99 & 24.43 & 23.84 \\ 18.88 & 20.09 & 18.97 & 19.92 & 21.24 & 20.27 & 19.70 & 20.68 & 20.95 \\ 13.43 & 13.87 & 13.07 & 14.44 & 15.15 & 14.20 & 13.97 & 14.58 & 14.13 \end{bmatrix}$ | $\begin{bmatrix} 4.28 & 8.62 & 11.76 & 0.46 & 3.43 & 7.05 & -0.90 & 0.13 & 2.52 \\ 8.52 & 16.03 & 21.88 & 1.49 & 6.46 & 13.16 & -2.07 & 0.43 & 5.53 \\ -1.98 & -3.77 & -3.37 & 0.36 & -1.19 & -1.98 & 2.68 & 1.78 & 0.47 \\ -7.44 & -12.73 & -13.18 & -1.24 & -6.06 & -9.47 & 3.27 & 1.34 & -3.06 \end{bmatrix}$ |
| | $\Sigma$ | $\begin{bmatrix} 10.12 & 8.07 & 5.32 & 3.97 \\ 8.07 & 26.56 & 16.33 & 10.51 \\ 5.32 & 16.33 & 23.70 & 11.04 \\ 3.97 & 10.51 & 11.04 & 12.46 \end{bmatrix}$ | $\begin{bmatrix} 17.43 & 24.37 & -13.00 & -27.12 \\ 24.37 & 50.79 & -25.58 & -53.50 \\ -13.00 & -25.58 & 44.79 & 53.03 \\ -27.12 & -53.50 & 53.03 & 98.70 \end{bmatrix}$ |
| | $\Psi$ | $\begin{bmatrix} 2.16 & 1.36 & 0.76 & 1.12 & 0.80 & 0.43 & 0.66 & 0.46 & 0.22 \\ 1.36 & 2.03 & 1.39 & 0.84 & 0.88 & 0.89 & 0.43 & 0.45 & 0.45 \\ 0.76 & 1.39 & 2.30 & 0.56 & 0.81 & 1.30 & 0.26 & 0.46 & 0.74 \\ 1.12 & 0.84 & 0.56 & 1.95 & 0.99 & 0.49 & 1.21 & 0.90 & 0.47 \\ 0.80 & 0.88 & 0.81 & 0.99 & 1.51 & 0.97 & 0.69 & 0.77 & 0.72 \\ 0.43 & 0.89 & 1.30 & 0.49 & 0.97 & 2.07 & 0.45 & 0.72 & 1.15 \\ 0.66 & 0.43 & 0.26 & 1.21 & 0.69 & 0.45 & 2.53 & 1.44 & 0.72 \\ 0.46 & 0.45 & 0.46 & 0.90 & 0.77 & 0.72 & 1.44 & 1.86 & 1.12 \\ 0.22 & 0.45 & 0.74 & 0.47 & 0.72 & 1.15 & 0.72 & 1.12 & 1.91 \end{bmatrix}$ | $\begin{bmatrix} 2.78 & 1.88 & 1.31 & 2.00 & 1.60 & 0.97 & 1.35 & 1.22 & 0.97 \\ 1.88 & 2.18 & 1.62 & 1.47 & 1.35 & 1.12 & 0.96 & 0.81 & 0.79 \\ 1.31 & 1.62 & 2.20 & 1.18 & 1.06 & 1.11 & 0.83 & 0.69 & 0.64 \\ 2.00 & 1.47 & 1.18 & 2.71 & 1.80 & 1.08 & 2.06 & 1.83 & 1.32 \\ 1.60 & 1.35 & 1.06 & 1.80 & 2.14 & 1.45 & 1.51 & 1.59 & 1.54 \\ 0.97 & 1.12 & 1.11 & 1.08 & 1.45 & 1.96 & 1.22 & 1.31 & 1.53 \\ 1.35 & 0.96 & 0.83 & 2.06 & 1.51 & 1.22 & 3.23 & 2.44 & 1.81 \\ 1.22 & 0.81 & 0.69 & 1.83 & 1.59 & 1.31 & 2.44 & 2.83 & 2.19 \\ 0.97 & 0.79 & 0.64 & 1.32 & 1.54 & 1.53 & 1.81 & 2.19 & 2.76 \end{bmatrix}$ |
| MVMMNE | M | $\begin{bmatrix} 55.37 & 54.60 & 54.73 & 54.53 & 54.28 & 54.58 & 54.48 & 54.69 & 54.81 \\ 77.59 & 76.71 & 76.64 & 76.07 & 75.70 & 75.73 & 75.58 & 75.60 & 75.71 \\ 93.17 & 92.58 & 92.79 & 92.33 & 91.34 & 91.72 & 92.10 & 91.85 & 91.06 \\ 77.99 & 77.82 & 78.22 & 77.26 & 76.82 & 77.26 & 77.62 & 77.13 & 76.99 \end{bmatrix}$ | $\begin{bmatrix} 46.82 & 44.03 & 42.84 & 48.46 & 46.39 & 44.35 & 49.94 & 48.95 & 47.67 \\ 35.60 & 31.34 & 29.34 & 38.63 & 34.79 & 31.45 & 41.74 & 39.62 & 37.01 \\ 114.94 & 116.50 & 115.98 & 114.24 & 115.51 & 116.47 & 112.39 & 112.94 & 113.59 \\ 122.24 & 125.79 & 125.21 & 119.77 & 123.59 & 125.29 & 116.45 & 117.85 & 119.93 \end{bmatrix}$ |
| | $\Lambda$ | $\begin{bmatrix} 8.68 & 9.19 & 8.69 & 9.09 & 9.06 & 8.47 & 8.51 & 8.16 & 8.11 \\ 18.38 & 19.16 & 18.43 & 19.59 & 20.12 & 19.29 & 19.22 & 19.57 & 19.10 \\ 15.14 & 16.08 & 15.14 & 15.94 & 16.99 & 16.19 & 15.77 & 16.56 & 16.78 \\ 10.77 & 11.08 & 10.42 & 11.56 & 12.12 & 11.34 & 11.16 & 11.66 & 11.32 \end{bmatrix}$ | $\begin{bmatrix} 2.26 & 4.61 & 6.29 & 0.41 & 1.99 & 3.93 & -0.15 & 0.40 & 1.68 \\ 4.53 & 8.51 & 11.64 & 1.06 & 3.70 & 7.32 & -0.46 & 0.83 & 3.51 \\ -1.23 & -2.21 & -2.02 & -0.16 & -1.03 & -1.50 & 0.84 & 0.38 & -0.34 \\ -4.12 & -6.89 & -7.20 & -1.14 & -3.72 & -5.61 & 0.84 & -0.19 & -2.51 \end{bmatrix}$ |
| | $\Sigma$ | $\begin{bmatrix} 10.55 & 8.53 & 5.66 & 4.22 \\ 8.53 & 27.63 & 17.04 & 10.98 \\ 5.66 & 17.04 & 24.61 & 11.50 \\ 4.22 & 10.98 & 11.50 & 12.93 \end{bmatrix}$ | $\begin{bmatrix} 18.13 & 25.28 & -13.55 & -28.27 \\ 25.28 & 52.87 & -26.70 & -55.89 \\ -13.55 & -26.70 & 47.29 & 55.88 \\ -28.27 & -55.89 & 55.88 & 103.84 \end{bmatrix}$ |
| | $\Psi$ | $\begin{bmatrix} 2.11 & 1.34 & 0.77 & 1.11 & 0.80 & 0.44 & 0.67 & 0.47 & 0.24 \\ 1.34 & 2.00 & 1.37 & 0.85 & 0.89 & 0.90 & 0.45 & 0.47 & 0.46 \\ 0.77 & 1.37 & 2.25 & 0.57 & 0.82 & 1.29 & 0.28 & 0.48 & 0.74 \\ 1.11 & 0.85 & 0.57 & 1.91 & 0.99 & 0.50 & 1.20 & 0.90 & 0.48 \\ 0.80 & 0.89 & 0.82 & 0.99 & 1.49 & 0.97 & 0.70 & 0.77 & 0.73 \\ 0.44 & 0.90 & 1.29 & 0.50 & 0.97 & 2.04 & 0.47 & 0.72 & 1.14 \\ 0.67 & 0.45 & 0.28 & 1.20 & 0.70 & 0.47 & 2.47 & 1.41 & 0.72 \\ 0.47 & 0.47 & 0.48 & 0.90 & 0.77 & 0.72 & 1.41 & 1.82 & 1.11 \\ 0.24 & 0.46 & 0.74 & 0.48 & 0.73 & 1.14 & 0.72 & 1.11 & 1.87 \end{bmatrix}$ | $\begin{bmatrix} 2.65 & 1.78 & 1.24 & 1.90 & 1.51 & 0.91 & 1.27 & 1.15 & 0.90 \\ 1.78 & 2.05 & 1.51 & 1.39 & 1.26 & 1.02 & 0.88 & 0.74 & 0.71 \\ 1.24 & 1.51 & 2.04 & 1.10 & 0.98 & 1.00 & 0.74 & 0.61 & 0.56 \\ 1.90 & 1.39 & 1.10 & 2.58 & 1.71 & 1.02 & 1.97 & 1.74 & 1.25 \\ 1.51 & 1.26 & 0.98 & 1.71 & 2.02 & 1.35 & 1.43 & 1.51 & 1.44 \\ 0.91 & 1.02 & 1.00 & 1.02 & 1.35 & 1.82 & 1.13 & 1.22 & 1.42 \\ 1.27 & 0.88 & 0.74 & 1.97 & 1.43 & 1.13 & 3.09 & 2.33 & 1.72 \\ 1.15 & 0.74 & 0.61 & 1.74 & 1.51 & 1.22 & 2.33 & 2.70 & 2.08 \\ 0.90 & 0.71 & 0.56 & 1.25 & 1.44 & 1.42 & 1.72 & 2.08 & 2.61 \end{bmatrix}$ |
| MVMMNW | M | $\begin{bmatrix} 51.52 & 50.57 & 50.93 & 50.53 & 50.32 & 50.89 & 50.74 & 51.09 & 51.27 \\ 70.14 & 68.86 & 69.10 & 68.14 & 67.51 & 67.91 & 67.93 & 67.72 & 68.00 \\ 87.15 & 86.03 & 86.52 & 85.90 & 84.42 & 85.04 & 85.75 & 85.13 & 84.26 \\ 73.70 & 73.23 & 73.86 & 72.62 & 71.90 & 72.58 & 73.06 & 72.37 & 72.40 \end{bmatrix}$ | $\begin{bmatrix} 45.01 & 39.48 & 36.28 & 49.35 & 45.19 & 40.75 & 51.86 & 50.09 & 47.16 \\ 32.00 & 22.74 & 17.18 & 39.81 & 32.33 & 24.59 & 45.62 & 41.55 & 35.42 \\ 114.99 & 117.62 & 116.95 & 112.69 & 114.92 & 116.43 & 109.29 & 110.53 & 111.96 \\ 124.78 & 131.89 & 131.66 & 118.19 & 125.29 & 129.24 & 111.68 & 114.63 & 119.67 \end{bmatrix}$ |
| | $\Lambda$ | $\begin{bmatrix} 13.21 & 13.93 & 13.16 & 13.79 & 13.73 & 12.81 & 12.91 & 12.39 & 12.27 \\ 27.21 & 28.46 & 27.37 & 29.00 & 29.83 & 28.56 & 28.32 & 28.93 & 28.26 \\ 22.30 & 23.84 & 22.57 & 23.58 & 25.19 & 24.10 & 23.31 & 24.52 & 24.85 \\ 15.88 & 16.51 & 15.57 & 17.08 & 17.95 & 16.88 & 16.57 & 17.31 & 16.76 \end{bmatrix}$ | $\begin{bmatrix} 4.77 & 10.74 & 15.06 & -0.57 & 3.74 & 8.84 & -2.43 & -0.86 & 2.57 \\ 9.54 & 20.06 & 27.91 & -0.13 & 7.21 & 16.62 & -5.09 & -1.29 & 5.99 \\ -1.51 & -3.90 & -3.50 & 1.62 & -0.51 & -1.71 & 4.62 & 3.27 & 1.51 \\ -7.81 & -15.23 & -16.00 & 0.52 & -6.36 & -11.22 & 6.57 & 3.55 & -2.64 \end{bmatrix}$ |

(*Continued*)

**Table 5.** (Continued)

| Dataset | Parameter | red soil | | | cotton crop | | |
|---|---|---|---|---|---|---|---|
| | Σ | $\begin{bmatrix} 10.04 & 7.89 & 5.17 & 3.86 \\ 7.89 & 26.42 & 16.21 & 10.42 \\ 5.17 & 16.21 & 23.60 & 10.97 \\ 3.86 & 10.42 & 10.97 & 12.41 \end{bmatrix}$ | | | $\begin{bmatrix} 17.55 & 24.60 & -13.16 & -27.40 \\ 24.60 & 51.26 & -25.85 & -54.03 \\ -13.16 & -25.85 & 44.80 & 53.22 \\ -27.40 & -54.03 & 53.22 & 99.23 \end{bmatrix}$ | | |
| | Ψ | $\begin{bmatrix} 2.14 & 1.34 & 0.73 & 1.10 & 0.77 & 0.40 & 0.64 & 0.43 & 0.19 \\ 1.34 & 2.00 & 1.36 & 0.82 & 0.85 & 0.86 & 0.41 & 0.42 & 0.42 \\ 0.73 & 1.36 & 2.27 & 0.53 & 0.78 & 1.27 & 0.23 & 0.44 & 0.71 \\ 1.10 & 0.82 & 0.53 & 1.93 & 0.97 & 0.46 & 1.19 & 0.88 & 0.44 \\ 0.77 & 0.85 & 0.78 & 0.97 & 1.48 & 0.95 & 0.67 & 0.74 & 0.70 \\ 0.40 & 0.86 & 1.27 & 0.46 & 0.95 & 2.05 & 0.43 & 0.69 & 1.13 \\ 0.64 & 0.41 & 0.23 & 1.19 & 0.67 & 0.43 & 2.52 & 1.42 & 0.70 \\ 0.43 & 0.42 & 0.44 & 0.88 & 0.74 & 0.69 & 1.42 & 1.83 & 1.10 \\ 0.19 & 0.42 & 0.71 & 0.44 & 0.70 & 1.13 & 0.70 & 1.10 & 1.89 \end{bmatrix}$ | | | $\begin{bmatrix} 2.82 & 1.94 & 1.39 & 2.02 & 1.63 & 1.02 & 1.37 & 1.24 & 1.00 \\ 1.94 & 2.24 & 1.69 & 1.53 & 1.39 & 1.16 & 1.02 & 0.86 & 0.83 \\ 1.39 & 1.69 & 2.28 & 1.26 & 1.12 & 1.16 & 0.91 & 0.77 & 0.71 \\ 2.02 & 1.53 & 1.26 & 2.70 & 1.82 & 1.13 & 2.05 & 1.82 & 1.34 \\ 1.63 & 1.39 & 1.12 & 1.82 & 2.16 & 1.48 & 1.52 & 1.61 & 1.56 \\ 1.02 & 1.16 & 1.16 & 1.13 & 1.48 & 1.99 & 1.26 & 1.35 & 1.56 \\ 1.37 & 1.02 & 0.91 & 2.05 & 1.52 & 1.26 & 3.19 & 2.42 & 1.81 \\ 1.24 & 0.86 & 0.77 & 1.82 & 1.61 & 1.35 & 2.42 & 2.82 & 2.20 \\ 1.00 & 0.83 & 0.71 & 1.34 & 1.56 & 1.56 & 1.81 & 2.20 & 2.78 \end{bmatrix}$ | | |

| | | red soil | | | cotton crop | | |
|---|---|---|---|---|---|---|---|
| Model | Criterion → | $\ell_{max}$ | AIC | BIC | $\ell_{max}$ | AIC | BIC |
| RMVSN | | -46110.78 | 92475.55 | 93000.49 | -24169.20 | 48592.41 | 49025.69 |
| MVMMNE | | -46167.80 | 92589.60 | 93114.54 | **-24137.68** | **48529.37** | **48962.65** |
| MVMMNW | | **-46079.34** | **92412.68** | **92937.62** | -24183.09 | 48620.18 | 49053.46 |

## 7 Conclusion

This paper has introduced a new family of matrix variate distributions whose component pdfs arise from the mean-mixture of matrix variate normal model. Some properties and characteristics as well as three special cases of the new model are derived. We have developed a computationally EM-based algorithm for calibrating the matrix type parameters to the data. It is shown that the MVMMN distribution is closed under the formation of marginal and conditional distributions and under affine transformation which make it flexible to use in the various fields of three-variate data analysis, such as multivariate time series, image processing and longitudinal data analysis. Simulation results show that the ML estimates obtained via the ECM algorithm are empirically consistent. Moreover, numerical results from application to real dataset reveal that the proposed model is well suited in dealing with the skewed matrix variate experimental data.

The utility of our current approach can be extended to accommodate censored data based on a recent work studied in the multivariate case by [31, 32]. It may also be interesting to propose a family of scale mixture of MVMMN distribution to deal with heavy tailed three-way data. Another possible extension of the work herein is to consider finite mixture model based on the MVMMN distribution as a promising tools in classification and clustering heterogeneous matrix-valued asymmetric data [19, 33]. It would be of interest the distributions of the associated eigenvalues of the quadratic form (Theorem 8; for the complex form) to compute the channel capacity in wireless communication systems, since experimental data do not follow necessarily a normal distribution (see [34, 35]). All computations were carried out by R language and the computer program is available from the first author upon request.

## Appendix A: Comparison of contour plots of the MMN and MVMN families

Fig 2 illustrate the contour plots of the bivariate rSN and bivariate exponentiated MMN (MMNE) distributions as special cases of MMN family as well as the contour plots of the bivariate generalized hyperbolic skew-*t* (GHST) and bivariate normal inverse Gaussian (NIG) distributions as special cases of MVMN family. *O*

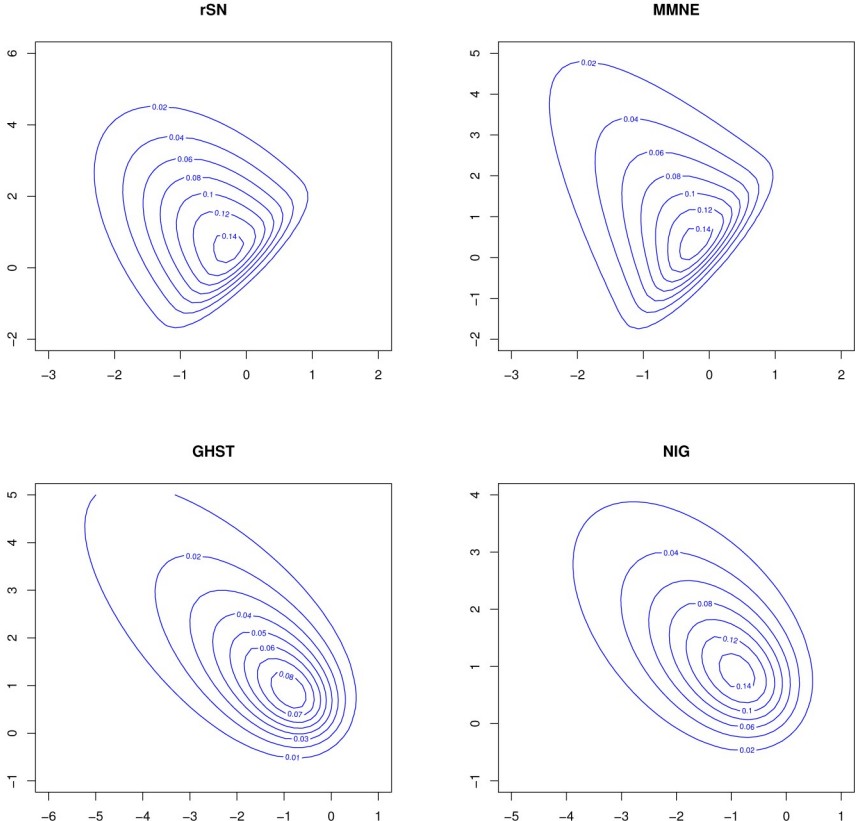

**Fig 2. Contour plots comparison of special cases of the MMN and MVMN families.**

## Acknowledgments

Our sincere thanks go to the anonymous referees and the Associate Editor, for their comments which led to a considerable improvement on an earlier version of this paper.

## Author Contributions

**Conceptualization:** Mehrdad Naderi.

**Methodology:** Mehrdad Naderi.

**Software:** Mehrdad Naderi.

**Writing – original draft:** Mehrdad Naderi.

**Writing – review & editing:** Andriette Bekker, Mohammad Arashi, Ahad Jamalizadeh.

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
