## [Decision Letter · Decision Letter 0]

11 Dec 2019

PONE-D-19-24001

A theoretical framework for Landsat data modeling based on the matrix variate mean-mixture of normal model

PLOS ONE

Dear Dr Naderi,

Thank you for submitting your manuscript to PLOS ONE. After careful consideration, we feel that it has merit but does not fully meet PLOS ONE’s publication criteria as it currently stands. Therefore, we invite you to submit a revised version of the manuscript that addresses the points raised during the review process.

We would appreciate receiving your revised manuscript by Jan 25 2020 11:59PM. To enhance the reproducibility of your results, we recommend that if applicable you deposit your laboratory protocols in protocols.io, where a protocol can be assigned its own identifier (DOI) such that it can be cited independently in the future. For instructions see: http://journals.plos.org/plosone/s/submission-guidelines#loc-laboratory-protocols

We look forward to receiving your revised manuscript.

Kind regards,

Daniel Capella Zanotta

Academic Editor

PLOS ONE

Journal Requirements:

'No'

Additional Editor Comments (if provided):

Dear author, we are now sending you the results of the reviewing by one referee. The remaing ones never materialized, so we decided, since it pointed minor revision, to proceed with this only one.

Please, improve the theoretical framework of the process putting most of the things in an easier way. Please, add coments clarifying each of the mathematical steps allowing future readers to better undertanding your intentions. As suggested by the reviewer, try to put auxiliar figures and schemes like flowcharts.

Reviewers' comments:

Reviewer's Responses to Questions

**Comments to the Author**

1. Is the manuscript technically sound, and do the data support the conclusions?

Reviewer #1: Yes

2. Has the statistical analysis been performed appropriately and rigorously? 

Reviewer #1: Yes

3. Have the authors made all data underlying the findings in their manuscript fully available?

Reviewer #1: Yes

4. Is the manuscript presented in an intelligible fashion and written in standard English?

Reviewer #1: Yes

5. Review Comments to the Author

Reviewer #1: The author's research is valuable and significant. In this paper, authors show a strong foundation in mathematics and physics, mathematical logic is rigorous. This research is innovative, however, the theoretical process would be easier to understand if more graphics were used.

6. PLOS authors have the option to publish the peer review history of their article (what does this mean?). If published, this will include your full peer review and any attached files.

Reviewer #1: No

---

## [Author Response · Author response to Decision Letter 0]

27 Jan 2020

We highly appreciate the reviewer for encouraging words and nice suggestions. The paper has been revised along the lines suggested.

---

## [Editor Report · Decision Letter 1]

10 Mar 2020

A theoretical framework for Landsat data modeling based on the matrix variate mean-mixture of normal model

PONE-D-19-24001R1

Dear Dr. Naderi,

We are pleased to inform you that your manuscript has been judged scientifically suitable for publication and will be formally accepted for publication once it complies with all outstanding technical requirements.

With kind regards,

Daniel Capella Zanotta

Academic Editor

PLOS ONE

Additional Editor Comments (optional):

Dear Dr. Naderi,

Given the delay presented by the referee, I checked your corrections by myself and I have found your paper ready for publication.

Please, follow the instructions above.

Reviewers' comments:

Made directly by academic editor.

---

## [Editor Report · Acceptance letter]

20 Mar 2020

PONE-D-19-24001R1 

A theoretical framework for Landsat data modeling based on the matrix variate mean-mixture of normal model 

Dear Dr. Naderi:

I am pleased to inform you that your manuscript has been deemed suitable for publication in PLOS ONE. Congratulations! Your manuscript is now with our production department. 

With kind regards,

on behalf of

Dr. Daniel Capella Zanotta 

Academic Editor

PLOS ONE